# Isotopic evidence for an intensified hydrological cycle in the Indian sector of the Southern Ocean

Camille Hayatte Akhoudas [1,2] ✉, Jean-Baptiste Sallée[3], Gilles Reverdin[3], F. Alexander Haumann [4,5,6], Etienne Pauthenet [7], Christopher C. Chapman [8], Félix Margirier [9], Claire Lo Monaco[3], Nicolas Metzl [3], Julie Meilland [10] & Christian Stranne [1,2]

The hydrological cycle is expected to intensify in a warming climate. However, observational evidence of such changes in the Southern Ocean is difficult to obtain due to sparse measurements and a complex superposition of changes in precipitation, sea ice, and glacial meltwater. Here we disentangle these signals using a dataset of salinity and seawater oxygen isotope observations collected in the Indian sector of the Southern Ocean. Our results show that the atmospheric water cycle has intensified in this region between 1993 and 2021, increasing the salinity in subtropical surface waters by $0.06 \pm 0.07$ g kg$^{-1}$ per decade, and decreasing the salinity in subpolar surface waters by $-0.02 \pm 0.01$ g kg$^{-1}$ per decade. The oxygen isotope data allow to discriminate the different freshwater processes showing that in the subpolar region, the freshening is largely driven by the increase in net precipitation (by a factor two) while the decrease in sea ice melt is largely balanced by the contribution of glacial meltwater at these latitudes. These changes extend the growing evidence for an acceleration of the hydrological cycle and a melting cryosphere that can be expected from global warming.

Over the past several decades, the Southern Ocean has experienced a complex array of hydrographic changes. These include large areas of the subpolar sector, south of the Antarctic Circumpolar Current (ACC), that have slightly cooled at the ocean surface[1–5], contrasting with a marked warming north of the ACC extending from the surface to at least 1000 m[4,6,7]. The temperature changes are accompanied with a near-surface freshening in the subpolar Southern Ocean and extending at depth into the Antarctic intermediate waters[5,8]. These patterns of changes have important consequences for regional and global climate. The large warming north of the ACC indicates the capacity of the

Southern Ocean to take up large amounts of excess heat associated with climate change and to store it in intermediate layers, thereby buffering global atmospheric warming[2,6,9]. Changes in salinity mirror shifts in the water cycle that strengthen the static stability of the subpolar upper ocean, with consequences for the large-scale circulation[10–12], sea ice regime[13], and surface subpolar ocean cooling[5,14,15]. However, despite their widespread importance, the processes that cause changes in Southern Ocean temperature and salinity remain not fully understood. A number of studies have suggested that the widespread subpolar Southern Ocean surface freshening is

[1]Department of Geological Sciences, Stockholm University, Stockholm, Sweden. [2]Bolin Centre for Climate Research, Stockholm University, Stockholm, Sweden. [3]CNRS/IRD/MNHN, LOCEAN, Sorbonne Université, Paris, France. [4]Alfred Wegener Institute, Helmholtz Centre for Polar and Marine Research, Bremerhaven, Germany. [5]Ludwig-Maximilians-University Munich, Munich, Germany. [6]Atmospheric and Oceanic Sciences Program, Princeton University, Princeton, NJ, USA. [7]LOPS, CNRS/IFREMER/IRD/UBO, Institut Universitaire Européen de la Mer, Plouzané, France. [8]CSIRO Environment, Earth Systems Science Program, Hobart, TAS, Australia. [9]School of Earth and Atmospheric Sciences, Georgia Institute of Technology, Atlanta, GA, USA. [10]MARUM, University of Bremen, Bremen, Germany. ✉e-mail: camille.akhoudas@geo.su.se

consistent with an amplification of the global hydrological cycle associated with an increase in net precipitation at high latitudes[6,16–18]. Other studies have proposed different mechanisms including intensified freshwater flux from increased ice shelf and iceberg melting[17,19,20], or from intensification of sea ice freshwater transport[21]. While climate models suggest that changes in Southern Ocean salinity over the past decades can be attributed in part to human-induced forcing[6,22], these simulations are known to not include a realistic representation of changes in meltwater from Antarctic ice shelves, as well as having questionable representation of Antarctic sea ice cover and trends, and a highly biased representation of net precipitation[2,23–27].

In this paper, we explore long-term observations of seawater oxygen isotope and salinity in surface waters of the Indian sector (between 40°E and 90°E, Fig. 1a, b) of the Southern Ocean to document regional ocean changes, and explore potential causes of surface freshening. Our analysis focuses only on the summer season, when

observations are available. The observational data span over 28 years, from 1993 to 2021, leveraging both the French Océan Indien Service d'Observations (OISO) program that started in 1998[28] and which provides a regional sampling of surface salinity and $\delta^{18}O$ (relative proportion of $^{18}O$ to $^{16}O$ referenced to Vienna Standard Mean Ocean Water) measurements; and additional hydrographic sections in the same sector of the Southern Ocean that span over the period 1993 to 1998. This dataset gives access to concurrent $\delta^{18}O$ observations, allowing for the investigation of long-term changes in the freshwater sources that have distinct $\delta^{18}O$ signatures. In the open ocean, away from the poles, the $\delta^{18}O$ of surface waters is predominantly controlled by atmospheric processes (similar to surface salinity)−increasing from evaporation and decreasing from precipitation. At the poles, however, $\delta^{18}O$ is also impacted by continental ice melt, which is very depleted in $\delta^{18}O$[29], while salinity is impacted by both continental ice melt and sea ice formation and melt. As a result, $\delta^{18}O$ is a useful tracer to

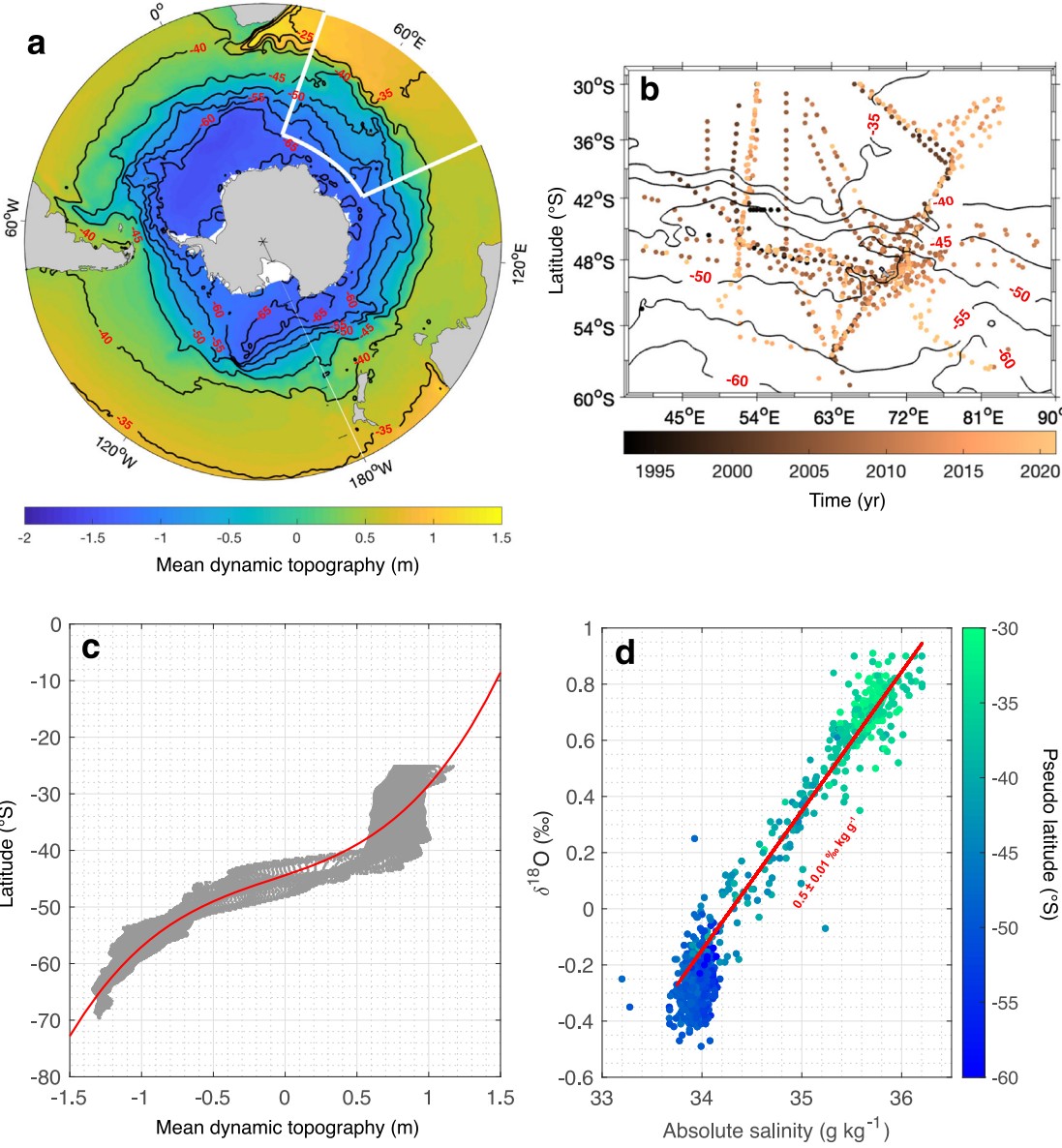

**Fig. 1 | Study area and overall presentation of salinity and oxygen isotope observations.** Mean dynamic topography[59] in color over the Southern Ocean (**a**). The white box represents the location of the ship-based observations obtained between 1993 and 2021 in color (**b**). Positions of the $\widetilde{lat}$ contours are plotted as black lines and labelled in red on both maps. (**c**) Ensemble of the zonal-mean dynamic topography as a function of latitude in grey. The best-fit for the distribution corresponds to a third-order polynomial curve as shown by the red line. (**d**) Salinity-$\delta^{18}O$ diagram of surface observations with $\widetilde{lat}$ in color. The salinity-$\delta^{18}O$ linear regression north of $\widetilde{lat}$ 46°S is shown as the red line (R-squared = 0.9) with its corresponding labelled slope−0.5 ± 0.01 ‰ kg g⁻¹.

## Results

### Regional surface trends

We compute the surface salinity-$\delta^{18}$O relationship, and meridional sections of zonally averaged values from a set of observations sampled from 1993 to 2021, in the region corresponding to 40–90°E and 30–60°S. Since the mean circulation structure of the region is not zonal, we work in a streamwise coordinate system, calculating all means, anomalies, and associated trends in bins of constant mean dynamic topography, which we translate back to geographical space (pseudo-latitude, $\widetilde{lat}$) for readability purposes (Fig. 1c, see Methods). The zonally averaged meridional profiles of salinity and $\delta^{18}$O have similar shapes: higher salinity and $\delta^{18}$O north of $\widetilde{lat}$ 46°S, and lower south of $\widetilde{lat}$ 46°S (Figs. 1d; 2a, b). The observed surface properties abruptly change across $\widetilde{lat}$ 46°S, identified as the northern boundary of the Subantarctic Front (SAF-N)[33] and associated with the ACC. North of the SAF-N (north of $\widetilde{lat}$ 46°S), the two properties actually follow a clear linear slope depending on the evaporation and precipitation characteristics of the region[34] (Fig. 1d). The salinity-$\delta^{18}$O slope in this region is 0.5 ± 0.01 ‰ kg g$^{-1}$, typical of slopes found in mid-latitude oceans[34]. In contrast, south of $\widetilde{lat}$ 46°S, the relationship between salinity and $\delta^{18}$O is much less clear (Fig. 1d) with small variations in salinity and comparatively much larger variability in $\delta^{18}$O (Fig. 1d). The large-scale meridional gradient of the isotopic composition of seawater is related to the gradual depletion in the $\delta^{18}$O of precipitation from the tropics toward higher latitude that influence surface ocean $\delta^{18}$O (Supplementary Fig. 1).

Based on the zonally averaged meridional profiles, we define the local spatial anomaly for each individual observation with respect to the climatologies in Fig. 2a,b. Then, we investigate whether salinity and $\delta^{18}$O anomalies tend to co-vary. The two distinct regimes in the subtropical and in the subpolar sectors—also referred to as the northern and the southern regions in the study—stand out clearly when computing correlation between salinity and $\delta^{18}$O anomalies. In the region north of $\widetilde{lat}$ 46°S, salinity and $\delta^{18}$O anomalies are strongly correlated (correlation coefficient mostly above 0.8; Fig. 2d). This high correlation is consistent with the observation of co-varying mean field (Fig. 1d), as a result of the two fields being mainly controlled by the same processes, evaporation and precipitation, along a fixed mixing line i.e., the amount of salinity changes for a given amount of $\delta^{18}$O changes, both set by evaporation and precipitation characteristics[35]. Salinity and $\delta^{18}$O of a water parcel is indeed dependent on evaporation and precipitation (and possibly other sources of meteoric waters and freshwater flux from sea ice) that the water parcel encountered during its lifetime, which tends to be advected by ocean currents. Conversely to the northern region, but still consistent with the mean fields, both parameters exhibit a substantially weaker relationship, inferred from low linear correlation coefficients, in the region south of $\widetilde{lat}$ 46°S (Fig. 2d). This poor correlation between anomalies of salinity and $\delta^{18}$O indicates that in addition to the local and non-local evaporation and precipitation, other processes are controlling surface salinity and $\delta^{18}$O. For example, while evaporation and precipitation have an impact on both parameters, $\delta^{18}$O is only marginally affected by sea ice formation and melt with a negligible fractionation factor on the order of 2‰[36,37]. Changes in sea ice formation and melt mostly impact the surface salinity. This feature of concomitant high salinity and low $\delta^{18}$O variations shows the influence of sea ice fluxes in surface waters advected from further south. The subpolar sector south of $\widetilde{lat}$ 46°S is thus imprinted by local evaporation and precipitation processes as well as non-local sea ice and meteoric (net precipitation and glacial meltwater) fluxes.

We now investigate long-term changes in salinity and $\delta^{18}$O. For this purpose, we first construct time series of annual medians of salinity and $\delta^{18}$O anomalies in the northern and southern regions (respectively north and south of $\widetilde{lat}$ 46°S; Fig. 3). Robust long-term trends are found in both sectors. By fitting a weighted linear regression model (see Methods), we find that the northern region experienced an increase of both surface salinity (Fig. 3a) and $\delta^{18}$O (Fig. 3b) from 1993 to 2021. In contrast, in the southern region both surface salinity (Fig. 3c) and $\delta^{18}$O (Fig. 3d) decreased over the same period of time. In order to gain further insight in the meridional variability of the trend, we also compute long-term trends in the same $\widetilde{lat}$ bins as Fig. 2. While we observe substantial variability along the meridional profile which is likely, at least partly, associated with observation limitations and uncertainties, the meridional profile of the trend clearly display two contrasting sectors on the northern and southern side of $\widetilde{lat}$ 46°S (Fig. 4). The magnitude is consistent with the trend computed in Fig. 3 (as represented by the grey shading in Fig. 4) with an increase of salinity and $\delta^{18}$O north of $\widetilde{lat}$ 46°S at rates of 0.06 ± 0.07 g kg$^{-1}$ per decade (Fig. 4a) and 0.03 ± 0.04‰ per decade (Fig. 4b), respectively, and a decrease south of $\widetilde{lat}$ 46°S at rates of −0.02 ± 0.01 g kg$^{-1}$ per decade (Fig. 4c) and -0.01 ± 0.02‰ per decade (Fig. 4d), respectively. While long-term trends of surface ocean $\delta^{18}$O in the Southern Ocean are limited to a small number of observations, trends in salinity can be investigated from much larger datasets[6,16,17,17–22]. This offers the opportunity to test the robustness of the estimated salinity trends from our dataset, which is more limited in terms of number of observations. Trends of mixed layer characteristics have recently been reported from a large combined database for the period 1970–2018[12]. Their independent analysis reports similar upper ocean salinity changes compared with what we infer from our dataset, with a similar meridional pattern and consistent magnitude (Supplementary Note 1 and Supplementary Fig. 2). When looking at the spatial pattern from an independent analysis based on this much larger dataset, salinity changes are found to be mostly consistent along streamlines of dynamic height, which comforts our approach to investigate changes in streamwise coordinates (Supplementary Fig. 2).

### Surface properties changes and associated forcing

The Southern Ocean is a data-sparse area making atmospheric reanalysis products generally the main tool for analysing observed surface ocean changes. However, there are inconsistencies among those products with a wide spread in precipitation minus evaporation (P-E)—positive P-E is a flux into the ocean—trends over the Southern Ocean (Supplementary Note 2 and Supplementary Figs. 5,6), which would introduce uncertainties when interpreting surface salinity changes observed in the Indian sector of the Southern Ocean. This caveat is a motivation to apply an independent approach by estimating the P-E changes from ocean properties and surface flux estimates of sea ice and glacial meltwater (see Methods).

Qualitatively, long-term salinification in the subtropics can be explained by a decrease in P-E, i.e., a decline of freshwater flux into the ocean. In contrast, long-term freshening in the subpolar sector could be explained by an increasing P-E, i.e., an amplification of freshwater flux into the ocean, a shift in sea ice regime, an increasing rate of ice sheet mass loss, or any combination of these. The weak correlation between salinity and $\delta^{18}$O in the southern region, and the strong correlation in the subtropics (Fig. 2d) suggest that the influence of sea ice (which tends to disrupt the correlation between salinity and $\delta^{18}$O) is limited to the subpolar region and does not considerably influence the subtropical region. The surface water influenced by sea ice in the subpolar sector is actually advected northward through meridional Ekman transport, but most of it subducts into intermediate and mode waters along the ACC fronts[38], and does not remain in the surface layer in the subtropical sector. Consequently, we aim to explain the observed changes in surface salinity and $\delta^{18}$O by a combination of changes in P-E, sea ice, and glacial meltwater inputs in the subpolar sector, and solely by changes in P-E in the subtropical sector

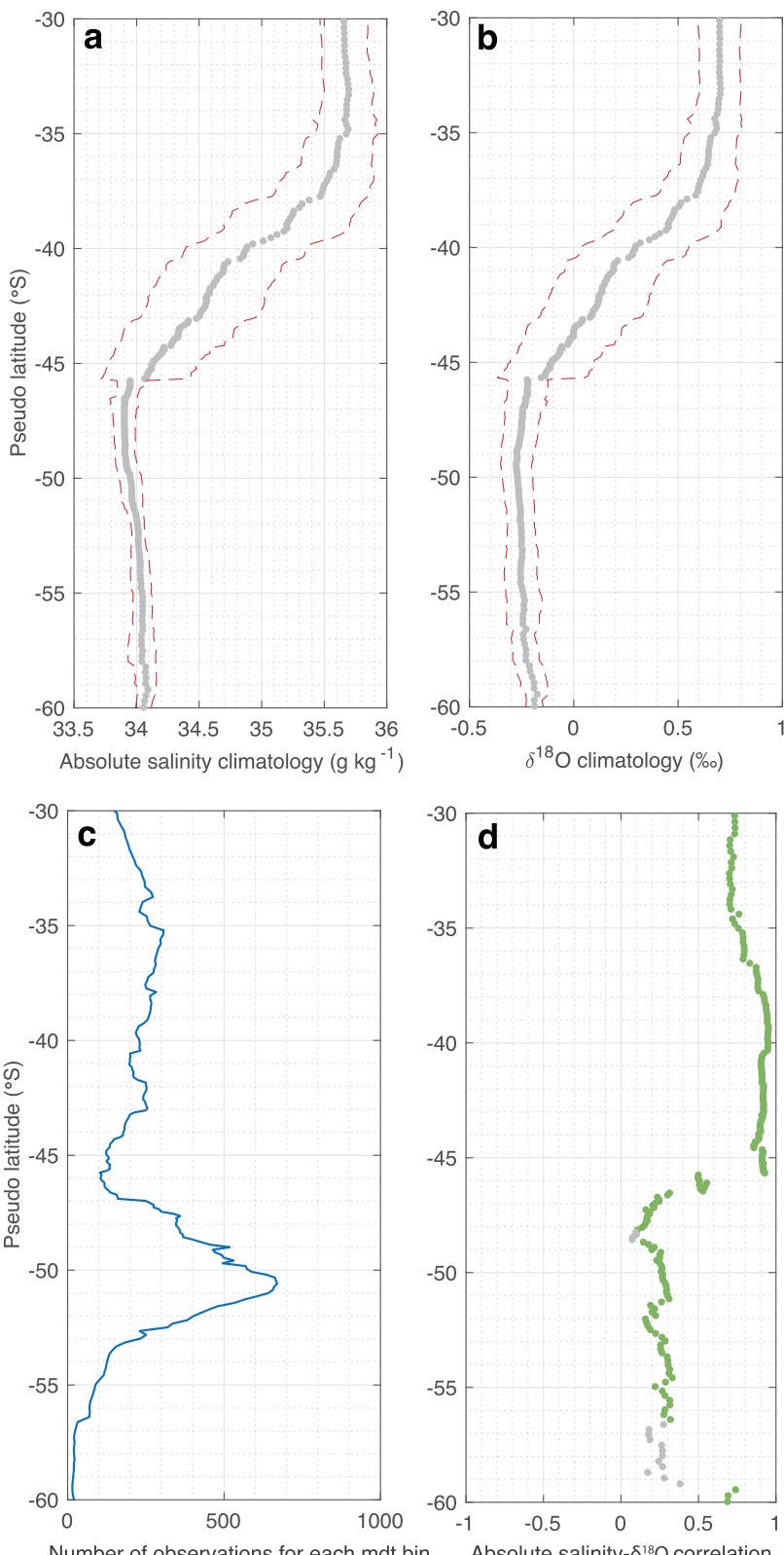

**Fig. 2 | Meridional profiles of summer surface salinity and oxygen isotopes in the Indian Southern Ocean.** Meridional profiles of zonally averaged salinity (**a**) and $\delta^{18}$O (**b**) and their standard errors in red dashed lines. Number of surface salinity and $\delta^{18}$O observations per bins of $\widetilde{lat}$ (**c**). Correlations between all salinity and $\delta^{18}$O observations in each bin of $\widetilde{lat}$ (**d**). Green dots correspond to significant correlation between the two variables with a $p$-value ≤ 0.05, conversely, grey dots are insignificant ($p$-value ≥ 0.05). Calculations are computed using an irregular mean dynamic topography grid with bin size ranging from 0.1 to 0.3 m.

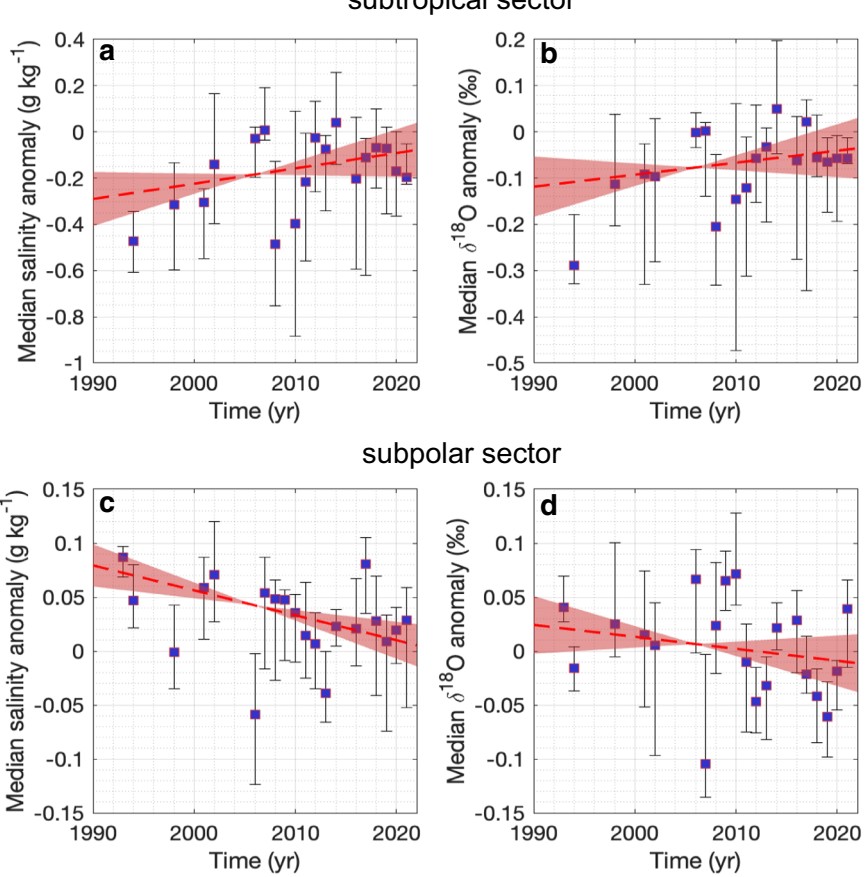

**Fig. 3 | Salinity and oxygen isotope anomalies time-series and trends in the subtropical and supolar sectors of the Indian Southern Ocean.** Median salinity (**a, c**) and $\delta^{18}$O (**b, d**) anomaly time series and associated trends for the subtropical (**a, b**) and subpolar (**c, d**) sectors. Each time series panel shows: the annual median anomaly (purple square from the regional salinity and $\delta^{18}$O climatology), computed for each individual observation; error bars shown in black referring to the 25th-75th percentile range; linear trends between 1993 and 2021 shown by the red dashed lines with their associated uncertainties, as the standard errors of the trends calculated using a bootstrap approach, in red shading.

(see Eqs. (4) and (5) in Methods). To do so, in the subtropical sector, we compute changes in P-E ($\Delta F_{P-E}$) from observed changes in salinity ($\Delta S$) and $\delta^{18}$O ($\Delta \delta^{18}$O):

$$\begin{cases} \Delta F_{P-E} = h\Delta S/(S_{P-E} - S_0), \\ \Delta F_{P-E} = h\Delta \delta^{18}O/(\delta^{18}O_{P-E} - \delta^{18}O_0). \end{cases}$$

where h is the depth of the mixed layer, $S_{P-E}$ and $\delta^{18}O_{P-E}$ are the salinity and $\delta^{18}$O of precipitation, $S_0$ and $\delta^{18}O_0$ the mean mixed layer salinity and $\delta^{18}$O. We repeat the computation of each of these two equations 50 000 times, but at each iteration we randomly perturb estimates of $\Delta S$, $\Delta \delta^{18}$O, $S_{P-E}$, $\delta^{18}O_{P-E}$, within their error ranges (see Table 1). We therefore obtain a set of 50 000 estimates of $\Delta F_{P-E}$ consistent with observed changes in $\Delta S$, and another set of 50 000 estimates of $\Delta F_{P-E}$ consistent with observed changes in $\delta^{18}O_{P-E}$. We then select all $\Delta F_{P-E}$ estimates that are consistent across the two set of solutions (within a range of ± 5 mm yr$^{-1}$ per decade). Our best estimate and associated error of $\Delta F_{P-E}$ is the mean ± standard deviation of these sub selected solutions. We also compute the corresponding best estimate and error of $\delta^{18}O_{P-E}$ with the same strategy (see Table 2). We find that our best estimate of changes in P-E resulting from this bootstrap exercise is a reduction of $-71 \pm 47$ mm yr$^{-1}$ per decade for precipitation with a mean isotopic composition of $-7 \pm 2$ ‰. We note that the impact of multidecadal changes in mixed layer depth is neglected in our calculation. ref. 12 showed that in this region the summer mixed layer depth has deepened at a rate of $2 \pm 4\%$ per decade, and this deepening represents

a third of the relative change in P-E based on the atmospheric reanalysis estimates (ERA5, JRA55 and GPCP/OAFlux).

In the subpolar sector, we repeat the exact same strategy, except that we have additional terms associated to the possible influence of sea ice and ice sheet (see Eqs. (4) and (5)). The bootstrap exercise is performed assuming a range for isotopic signatures of the Antarctic glacial meltwater[29,39] and sea ice[40] (see Table 1). We start by searching for solutions by assuming no changes have occurred in P-E, to test the hypothesis that observed changes could be solely driven by sea ice regime and glacial meltwater. No consistent solutions are found across Eqs. (4) and (5) under this assumption; so we conclude that changes in P-E must be involved in the observed surface ocean trends. When allowing for all forcing perturbations, i.e., P-E, glacial melt and sea ice, and selecting all consistent solutions across Eqs. (4) and (5), we find that the best estimate of P-E changes is an increase of $33 \pm 11$ mm yr$^{-1}$ per decade, with a mean precipitation $\delta^{18}$O of $-11 \pm 3$‰. In this solution, glacial melt changes by $11 \pm 4$ mm yr$^{-1}$ per decade, with a mean ice sheet $\delta^{18}$O of $-25 \pm 3$‰, and sea ice melt decreases by $-12 \pm 2$ mm yr$^{-1}$ per decade. An important result here is that ice sheet mass loss has a limited effect on salinity and $\delta^{18}$O trends compared to P-E freshwater flux. Our computed equivalent salinity trend induced by P-E amounts to $-0.02 \pm 0.008$ g kg$^{-1}$ per decade compensated by $0.008 \pm 0.002$ g kg$^{-1}$ per decade induced by sea ice and enhanced by $-0.008 \pm 0.003$ g kg$^{-1}$ per decade induced by glacial melt to sum up the total salinity trend observed in the subpolar sector. We note that we tested the extreme case where we assume that ice sheet mass loss had no impact on

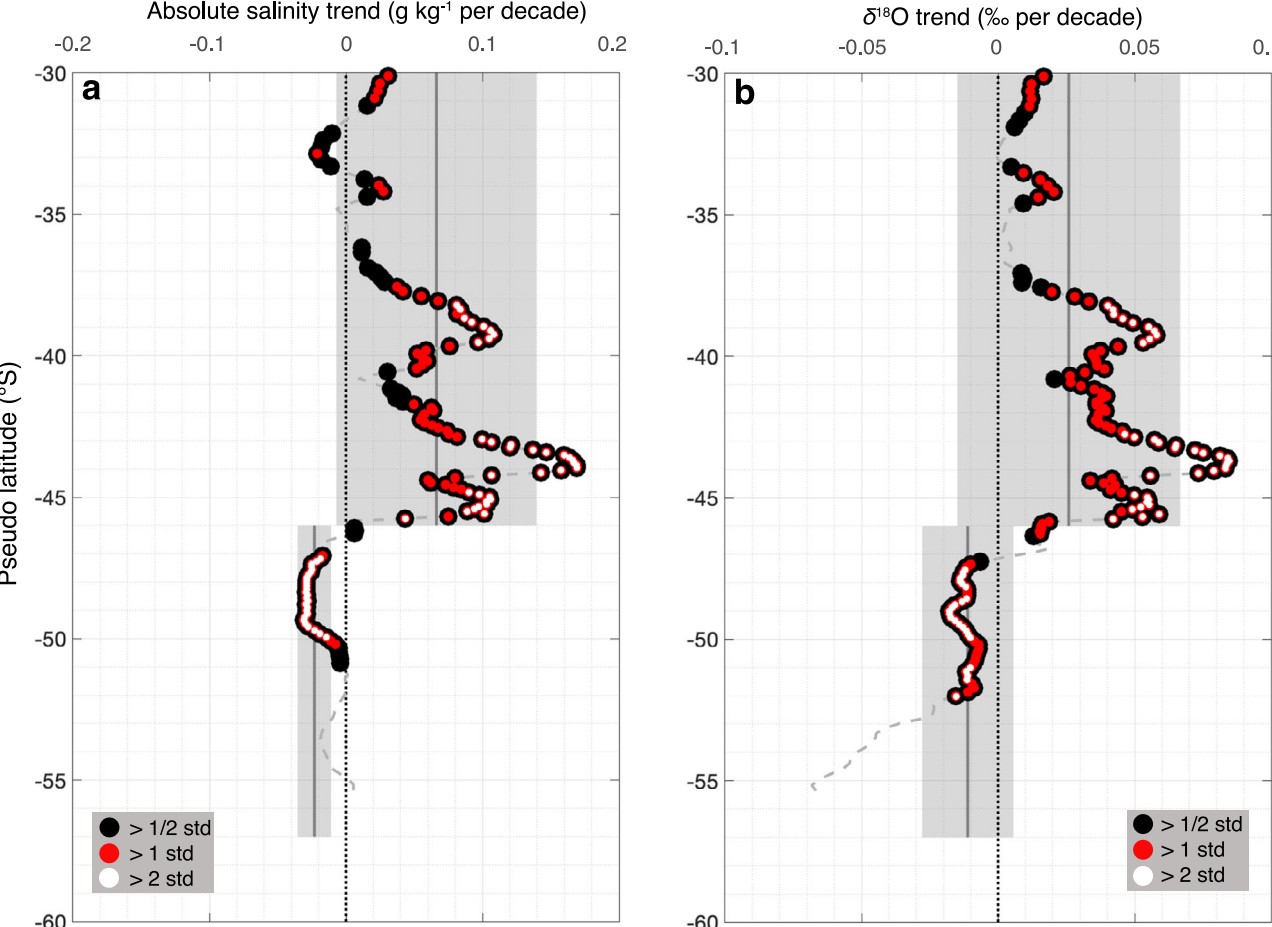

**Fig. 4 | Meridional profiles of salinity and oxygen isotope trends.** Meridional profiles of zonally averaged salinity (**a**) and $\delta^{18}$O (**b**) trends (grey dashed line). The black dots represent the trends larger than half their estimated standard errors (>1/2 std), the red dots the trends larger than their estimated standard errors (>1 std) and white dots the trends larger than double their estimated standard errors (>2 std). Trends are computed using an irregular mean dynamic topography grid with bin size ranging from 0.1 to 0.3 m. The grey lines show the mean salinity and $\delta^{18}$O trends calculated from linear regression models in each region shown in Fig. 3, and the grey shadings are their associated uncertainties estimated as the standard errors of the trends calculated using a bootstrap approach (see Fig. 3).

observed changes, and we are still able to find a plausible solution with a change in P-E freshwater flux slightly higher, corresponding to $44 \pm 10$ mm yr$^{-1}$ per decade. The region south of *lat* 52°S contains a much smaller amount of data than further north (Fig. 2c) and is associated with a different magnitude of salinity and $\delta^{18}$O trends; $-0.01 \pm 0.007$ g kg$^{-1}$ per decade (Fig. 4a) and $-0.04 \pm 0.02$‰ per decade (Fig. 4b), respectively. Even if these changes are computed in a shorter period of time, between 2001 and 2021 (Supplementary Fig. 7), the $\delta^{18}$O signal is considerably stronger. A significant decrease in $\delta^{18}$O coupled with a slight decrease in salinity is consistent with a decline in sea ice melt (Supplementary Note 2 and Supplementary Figs. 3,4) that is somewhat counteracting a higher decrease in salinity due to glacial meltwater and P-E increase.

## Discussion

Long-term salinity observations in the Southern Ocean have revealed changes over the past decades, which are among the most pronounced in the global ocean[6,8,16,18]. However, the source of these changes has remained a conundrum. Salinity observations alone are not able to delineate the different elements involved in the freshwater cycle in the Southern Ocean. Here, in addition to salinity, we use $\delta^{18}$O observations and focus on surface water properties in the Indian sector of the Southern Ocean between 1993 and 2021 during summer months (December to February). We investigate the changes in two sectors, south and north of *lat* 46°S, which corresponds to the northern branch of Subantarctic Front (SAF-N) of the ACC[33], separating Antarctic

Surface Waters in the south from Subtropical Surface Waters in the north. We attempt to minimize the aliasing of spatial variability by splitting measurements based on their respective dynamical region, as estimated by the climatological mean dynamic topography. We report a multidecadal salinification in the northern region of $0.06 \pm 0.07$ g kg$^{-1}$ per decade, associated with an increasing $\delta^{18}$O of $0.03 \pm 0.04$‰ per decade from 1993 to 2021. In contrast, the southern sector is freshened by $-0.02 \pm 0.01$ g kg$^{-1}$ per decade, and $\delta^{18}$O decreased at a rate of $-0.01 \pm 0.02$‰ per decade from 1993 to 2021. These results are consistent with the latest IPCC report that assessed with high confidence that human-induced climate change has driven detectable changes in the global water cycle since the mid-20th century, with enhanced contrasts in P-E patterns over the oceans, which are projected to continue in a warmer world[41]. We demonstrate that the observed surface ocean changes result from an acceleration of the hydrological cycle between 1993 and 2021, with a decrease in P-E of $-71 \pm 47$ mm yr$^{-1}$ per decade in the subtropical sector, contrasted by an increase in P-E of $33 \pm 11$ mm yr$^{-1}$ per decade in the subpolar sector. Our multidecadal trend estimates are based on a mean isotopic composition of precipitation of $-7 \pm 2$‰ in the northern sector and $-11 \pm 3$‰ in the southern sector, consistent with previously reported estimates and their associated errors[42] reported in Table 1 and taken into account in the equation solutions.

It has been reported that the increased glacial meltwater input due to ice shelf thinning[43,44] and enhanced flux at the grounding line[45]

**Table 1 | A priori range of values used to seek for solutions of the system of Eqs. (4) and (5)**

| | North of $\widetilde{lat}$ 46°S (Subtropical) | | | South of $\widetilde{lat}$ 46°S (Subpolar) | | |
|---|---|---|---|---|---|---|
| | S (g kg⁻¹) | δ¹⁸O (‰) | FWF (mm/yr/dec) | S (g kg⁻¹) | δ¹⁸O (‰) | FWF (mm/yr/dec) |
| Mixed layer | 35.20 ± 0.65 | 0.45 ± 0.33 | – | 33.95 ± 0.09 | −0.26 ± 0.08 | – |
| Sea ice | – | – | – | 4 ± 4 | 1 ± 1 | −12 ± 4 |
| Ice sheet | – | – | – | 0 | −30 ± 10 | 11 ± 6 |
| P-E | 0 | −5 ± 5 | – | 0 | −10 ± 5 | – |

The mixed layer properties are estimated as the mean and standard deviation of the observations in the corresponding regions from a climatology[11]. Ranges of plausible values for salinity (S) and δ¹⁸O characteristics are assessed based on available literature[29,34,39,40,42]. Similarly, ranges of plausible 1993-2021 changes in freshwater flux (FWF) for each component are estimated based on the relevant literature[21,46] (see Methods).

**Table 2 | Solutions of the system of Eqs. (4) and (5), run with the a priori provided in Table 1**

| | North of $\widetilde{lat}$ 46°S (Subtropical) | | | South of $\widetilde{lat}$ 46°S (Subpolar) | | |
|---|---|---|---|---|---|---|
| | S (g kg⁻¹) | δ¹⁸O (‰) | FWF (mm/yr/dec) | S (g kg⁻¹) | δ¹⁸O (‰) | FWF (mm/yr/dec) |
| Sea ice | – | – | – | 2 ± 1 | 0.5 ± 0.3 | −12 ± 2 |
| Ice sheet | – | – | – | 0 | −25 ± 3 | 11 ± 4 |
| P-E | 0 | −7 ± 2 | −71 ± 47 | 0 | −11 ± 3 | 33 ± 11 |

The only term that is unconstrained by an a priori is the change in freshwater flux (FWF) from P-E. Other terms are strongly constrained by a priori from existing literature, and we here provide the optimised best estimate allowing to find consistent solutions across Eqs. (4) and (5).

mostly drive changes on the continental shelf[5]. We find that the enhanced glacial meltwater discharge from the Antarctic ice shelves only has a very limited influence (inducing a salinity trend of −0.008 ± 0.003 g kg⁻¹ per decade of the total observed salinity change) on the open ocean salinity changes in the Indian sector of the Southern Ocean between 1993 and 2021. This result is consistent with refs. 5,17, stating that glacial freshwater flux increase is insufficient to explain the historical changes observed in the open ocean, away from the continental shelf. We note that as ice sheet mass loss is projected to continue in the future[46], the signature on surface ocean salinity and δ¹⁸O might become more pronounced. In contrast, sea ice decline (implying less formation and volume) induces a positive salinity trend (0.008 ± 0.002 g kg⁻¹ per decade) in the northern part of the sea ice sector and further north in the subpolar Indian sector of the Southern Ocean. This positive trend is compensated and surpassed by a negative trend induced by P-E (−0.02 ± 0.008 g kg⁻¹ per decade), leading to net surface ocean freshening. This feature is consistent with estimates from ref. 21 for this region, showing a decline in the northward sea ice transport in the Indian sector of the Southern Ocean and upstream of this region. The ocean's salinity response to this signal in a model[5] is a clear salinification, consistent with the slight negative salinity trend accompanied by the strong negative δ¹⁸O trend south of $\widetilde{lat}$ 52°S. Despite large regional variations, the total Antarctic sea ice cover slightly increased since the late 1970s[47] and sharply decreased in 2016[48], likely affecting the salinity anomalies towards the end of our time series (Fig. 3c). This sea ice decline, which continued after 2016, might become more prominent in the future, but should be covered by the relatively loose range of the estimate and its associated error of 33% applied in this study.

Our study quantifies the contribution of changes in freshwater forcing that influence surface water properties in the Southern Ocean directly from observations. Here we analyze the ocean response to both local and non-local freshwater forcing perturbations as signal from sea ice and meteoric waters is advected northward. Furthermore, the process of northward freshwater transport by sea ice contribute also to non-local contribution of meteoric waters through snow accumulated on top of sea ice. The results have important implications. First, we demonstrate the strength of the OISO long-term monitoring program recording concomitantly δ¹⁸O and hydrographic properties to disentangle complex processes currently at play in polar oceans. Second, our findings suggest only a limited influence of an

increase in Antarctic ice discharge on the surface water properties in the subpolar ocean, away from the continent. This is corroborated by the observational evidence of the depleted δ¹⁸O signal near the Antarctic coast[39,49], the increased glacial meltwater mostly affecting the waters along the continental shelf. The weak response to the open ocean to this coastal imprint is likely related to the significant mixing with higher δ¹⁸O waters across the shelf break where the depleted δ¹⁸O signal of waters flowing northward rapidly decreases[50]. However, this signal may become more prominent[23] in the future, as Antarctic ice discharge is projected to accelerate[51]. Our approach and continued monitoring of these surface ocean processes can be used as an early warning system to detect when and how ongoing ice sheet mass loss will start to impact ocean characteristics and the associated circulation, with important consequences and feedback for the global climate, and the ice sheet mass loss itself[23,52,53]. Third, providing a quantification of the role of forcings onto observed subpolar freshening and subtropical salinification opens the door to better gauge the role of human activities on ocean characteristics, and offers a framework and target for future climate model development to ensure that the critical processes in the polar regions are correctly represented. Finally, our results provide the quantifiable evidence of the contribution of different freshwater fluxes in the Indian sector of the Southern Ocean (40–90°E and 30–60°S). The use of the approach here whose strength comes from the concurrent observations of oxygen isotopes and salinity to disentangle the processes at play in surface salinity changes should be replicated in other regions of the Southern Ocean to address the spatially complex pattern of changes in surface water fluxes and their imprint on the ocean[5,54]. The different responses of local regime to direct or remote forcing are important to consider in order to investigate surface changes in the Southern Ocean as a whole.

## Methods

### Linking changes of freshwater forcing to changes of surface ocean properties

To assess how the changing atmospheric freshwater flux affected the surface ocean properties in the Indian sector of the Southern Ocean, we compute a possible range of P-E changes from salinity, δ¹⁸O observed trends and the best estimate of glacial meltwater and sea ice changes (Table 1). This approach enables us to overcome the inhomogeneities among reanalysis products (Supplementary Table 1). The

salinity budget of the mixed layer[55] can be written as:

$$h\frac{dS}{dt} = FWF(S_{FWF} - S_0) + Entr + Diff, \tag{1}$$

where h is the depth of the mixed layer; $S_0$ the mean mixed layer salinity; $FWF$ is a water flux at the ocean surface with a salinity $S_{FWF}$ in unit m s$^{-1}$ (we will hereafter refer to freshwater fluxes as $FWF$, although we note that these fluxes can have a positive salinity $S_{FWF}$); $Entr$ corresponds to the entrained salinity flux at the base of the mixed layer; and $Diff$ is the diffusive salinity flux across the mixed layer boundary, which can involve either vertical or horizontal mixing. Integrating the salinity budget over an entire seasonal cycle, and assuming that entrainment and diffusion have not changed, the long-term salinity change ΔS can be written as:

$$h\Delta S = (S_{FWF} - S_0)\Delta \int FWF, \tag{2}$$

where $\int FWF$ is the integrated freshwater flux over one year, and $\Delta \int FWF$ is its long-term change. Decomposing the freshwater component in P-E flux ($\int FWF = F_{P-E}$), sea ice induced flux ($\int FWF = F_{SI}$), and glacial meltwater flux ($\int FWF = F_{IS}$) gives:

$$h\Delta S = (S_{P-E} - S_0)\Delta F_{P-E} + (S_{SI} - S_0)\Delta F_{SI} + (S_{IS} - S_0)\Delta F_{IS}, \tag{3}$$

which we can rewrite as:

$$\Delta F_{P-E} = \frac{h\Delta S - (S_{SI} - S_0)\Delta F_{SI} - (S_{IS} - S_0)\Delta F_{IS}}{(S_{P-E} - S_0)} \tag{4}$$

Repeating the same exercise but for a $\delta^{18}$O budget integrated over the mixed layer leads us similarly to:

$$\Delta F_{P-E} = \frac{h\Delta\delta^{18}O - (\delta^{18}O_{SI} - \delta^{18}O_0)\Delta F_{SI} - (\delta^{18}O_{IS} - \delta^{18}O_0)\Delta F_{IS}}{(\delta^{18}O_{P-E} - \delta^{18}O_0)} \tag{5}$$

Here we use Eqs. (4) and (5) to determine long-term changes in P-E consistent with our best estimates of the other terms of the equations. In these equations, horizontal advection is neglected, as the coordinate system that we use follows the climatological mean position of the ACC fronts. Multidecadal changes in horizontal advection could be induced by, for example, a meridional displacement of the ACC fronts. However, recent assessments conclude that large meridional shifts of Southern Ocean fronts are unlikely over the past decades[2,46,56]. Any small potential southward shift of fronts would in fact lead to a signal of opposite sign than what we observe, i.e., higher salinities towards the south, and are thus unlikely to affect the overall conclusion of our study. In addition, the possibility of a slight change in the fronts position or any other approximation made in these equations are, to some extent, taken into consideration by repeating 50 000 times the computation of Eqs. (4) and (5), but varying all terms within their uncertainties. We then retain all solutions that provide consistent estimates of $F_{P-E}$ when computed by Eq. (4) compared to when computed by Eq. (5) (within a range of ±5 mm yr$^{-1}$ per decade). The uncertainties in $\Delta\delta^{18}O$ and $\Delta S$ are inferred from the standard errors of the computed trends, and all other terms and associated uncertainties are summarized in Table 2.

## Long-term $\delta^{18}$O and salinity observations
In this study, we use surface observations of oxygen-18 isotopic composition ($\delta^{18}$O) and salinity (expressed as absolute salinity following TEOS-10), from underway sampling at a depth of about 5–7 m from the sea surface and from stations sampling between a depth of 0 and 50 m, in the Indian sector of the Southern Ocean (40°E–90°E and 30°S–60°S,

Fig. 1a). These observations are from oceanographic research cruises undertaken during austral summer (December to February) from 1993 to 2021 (Fig. 1b). We note here that the depth of the measurements within the upper 50 meters does not influence the mean salinity and $\delta^{18}$O surface fields (Supplementary Fig. 8). Ship-based data used in this study are stored and distributed through the OISO database, for those sampled under the OISO program from 1998 to 2021[28], and through the GISS (Goddard Institute for Space Studies) database for those from 1993 to 1998[34]. All data flagged as "bad" or "probably bad" are discarded. Most of the observations were processed at the Laboratoire d'Océanographie et du Climat Expériments et Approches Numeriques (LOCEAN) in Paris, France. The seawater samples were analyzed using two different instrumental methods; (i) most data prior to 2010 were analyzed using isotope ratio mass spectrometry (Isoprime IRMS) coupled with a Multiprep system (dual inlet method), (ii) whereas most data since 2010 (and a few earlier data) were obtained by cavity ring down spectroscopy (CRDS) using a Picarro instrument (model L2130-i Isotopic H$_2$O) to measure for $\delta^{18}$O. Seawater samples were analyzed directly with the CRDS system using a stainless-steel liner from Picarro that is inserted in the injection port to avoid salt accumulation in the vaporizer. The use of a liner has the advantage of preserving the accuracy of the seawater isotope analyses as compared to direct injections into the vaporizer and the mesh traps about 80% of the seawater salt[57]. Once the database was calibrated, it was then corrected for the analyzer-dependent "sea salt effect"[57]. The $\delta^{18}$O database is corrected so that deep water properties, which should be stable over the time period considered in this study, are constant across the different datasets. Typically, we focused over a range of neutral densities spanning over 28.15 and 28.3 kg m$^{-3}\gamma_n$ which corresponds to the old and stable Circumpolar Deep Water and encompasses water masses with typical timescales of ventilation of the order of hundred years. This approach allowed us to correct for nonphysical offsets coming from different analytical techniques, different instruments and changes in standards. $\delta^{18}$O measurements uncertainty is on the order of 0.06‰ including instrumental and systematic errors associated with combining different datasets. In addition, we use concomitant salinity measurements from samples analyzed on a salinometer, calibrated CTD (Conductivity Temperature Depth) casts, and calibrated thermosalinograph data.

## Streamwise coordinate system
In this study, all climatological means, associated anomalies, and trends are computed within dynamically consistent regions with regards to the mean position of fronts, using a streamwise coordinate system. This streamwise coordinate system enables us to properly investigate ocean properties in this meandering region of the Southern Ocean, but neglects any potential meridional advection into the mixed layer. Because the geostrophic circulation follows contours of dynamic topography, we chose to work in bins of constant mean dynamic topography rather than bins of latitude[58]. In practice, all individual observations are ascribed a value of mean dynamic topography using the widely used product distributed by AVISO[59]. We then chose an irregular mean dynamic topography grid with bin size ranging from 0.1 to 0.3 m. Finer grids are used within the ACC since horizontal gradients of dynamic topography are larger there. Climatological means, anomalies, and trends are all performed in this streamwise coordinate system. For readability purpose, the results are shown in latitude space, referred to pseudo latitude ($\widetilde{lat}$), computed as the mean latitude in each mean dynamic topography bin (Fig. 1c).

## Trends and associated errors
Trends are estimated by fitting linear regression models with their associated standard errors. The confidence level is established by computing the standard error of the regression. For trends performed in streamwise coordinate bins, we indicate trend values that are (i) less than half the computed standard errors, (ii) between half and one

standard errors, (iii) between one and two standard errors, and (iv) more than two standard errors, to estimate the trend robustness with regards to the standard error. For trends produced in large sectors (north and south of $\widetilde{lat}$ 46°S), (i) we first compute local anomalies from the zonal-mean meridional profile by removing the mean of all observations in the streamwise coordinate bins; (ii) we then compute the annual 25, 50, 75 percentiles of anomalies in the large sector; (iii) finally, we fit a weighted linear regression model to the annual medians, weighted by $w = 1/IQR^2$, where IQR is the interquartile range (percentile 75 minus percentile 25), which provides an estimate of the long-term trend. Its associated uncertainty is then estimated as the standard error of the trend calculated using a bootstrap approach (repeated 50 000 times taking 80% of the values randomly).

### Changes in freshwater forcing in the Southern Ocean

Sea ice. The freshwater flux induced by the sea ice seasonal cycle and its interannual changes are obtained over the period 1993–2008 from ref. 21, in which sea ice freshwater flux is obtained by combining sea ice concentration, drift and thickness, coupled with a mass balance approach to determine the volume divergence and local change in sea ice (see their methods section for further details). The sea ice freshwater flux trends inferred from these published estimates in our region of interest, between 40°E and 90°E, is −31 ± 6 mm yr$^{-1}$ per decade corresponding to the mean over the entire sector, i.e. it tends to increase surface ocean salinity. The mean sea ice freshwater flux trend over the latitude band we are considering as the subpolar sector in this study, between 60°S and 46°S, correspond to −12 ± 4 mm yr$^{-1}$ per decade.

Ice sheet. Continental ice meltwater changes can come from either increased ice discharge at the Antarctic grounding lines, or from ice shelf thinning. Here we use the assessment of the latest IPCC report to estimate these two components[46]. In accordance with this assessment, the change in ice discharge is approximately equivalent to the change in mass balance. The Antarctic Ice Sheet mass loss rate was 49 ± 51 Gt yr$^{-1}$ from 1992 to 1999, 70 ± 59 Gt yr$^{-1}$ from 2000 to 2009, and 148 ± 54 Gt yr$^{-1}$ from 2010 to 2016[46]. In consequence, assuming no change from 1992 to 1993, and that the rate of change was the same from 2016 to 2021 as from 2005 to 2013 (10 ± 14 Gt yr$^{-2}$), we infer that the ice discharge increased by a total of 317 ± 118 Gt yr$^{-1}$ from 1993 to 2021, which translates into a mean acceleration of ice discharge corresponding to 113 ± 42 Gt yr$^{-1}$ per decade during this period. The latest IPCC assessment also reports estimates of basal melt related to ice shelf thinning with, however, substantial spatio-temporal variability[44,46]. They estimated an increase from 1100 ± 150 Gt yr$^{-1}$ to 1570 ± 140 Gt yr$^{-1}$ from the early 1990s to the late 2000s, but a decrease from 1570 ± 140 Gt yr$^{-1}$ to 1160 ± 150 Gt yr$^{-1}$ in the late 2010s. The reconstruction of time-varying total glacial freshwater fluxes from the Antarctic Ice Sheet is a challenge when considering changes between the early 1990s and the early 2020s only, with evidence for significant interannual variability[60] and a reduction in melt rates in the last decade[44]. Based on these published results, we consider that the best estimate of plausible changes from 1993 to 2021 due to ice shelf thinning ranges between 20 and 200 Gt yr$^{-1}$ per decade. In addition to these two components, ice mass loss associated with ice front retreat has been recently estimated as approximately equal to net ice mass loss due to ice shelf thinning[61]. Our best estimate of changes in total (grounding line plus double ice shelf thinning) Antarctic glacial meltwater input therefore ranges from about 150 to 550 Gt yr$^{-1}$ per decade from 1993 to 2021, consistent with the estimate discussed in ref. 62. Assuming that this glacial meltwater is evenly spread within the mixed layer over the entire subpolar region (south of the mean dynamic topography contour of −0.2 m; corresponding to $\widetilde{lat}$ 46°S; area of ~3.5 10$^7$ km$^2$), it converts into a freshwater flux trend ranging between 5 and 17 mm yr$^{-1}$ per decade from 1993 to 2021. However, we note that those changes in glacial meltwater flux are likely not evenly spread within the mixed layer over the entire subpolar region, and most likely have an impact to a greater extent in the coastal Antarctic region

than in the northern edge of the subpolar region of the Indian sector of the Southern Ocean. We therefore consider this rate of changes as an upper range estimate for the region considered in the present study.

Precipitation and Evaporation. As a result of the sparseness of in situ observations, atmospheric reanalyses have been widely used as a tool to report P-E over the Southern Ocean—the sign convention used here is that positive values of P-E refer to precipitation dominating over evaporation. However, long-term trends in P-E are highly uncertain due to the lack of measurements both spatially and temporally in the Southern Ocean in order to validate satellite products and modeled precipitation. Cautious use of reanalyses for climate change evaluation is thus needed as inhomogeneities between different products have been shown[63] and as we can observe between three different global atmospheric reanalysis products, ERA5[64], JRA55[65] and GPCP coupled with OAFlux[66,67], in our region of interest and at the circumpolar scale (Supplementary Note 2, Supplementary Figs. 3–6 and Supplementary Table 1).

## Data availability

Datasets for this study are available in the in-text data citation references and the associated repositories:

The salinity and $\delta^{18}$O data by refs. 28,34 (https://www.seanoe.org/data/00600/71186/and https://data.giss.nasa.gov/o18data/).

The salinity data by ref. 12 (https://github.com/jbsallee-ocean/GlobalMLDchange/tree/main/Databases).

The sea ice freshwater flux data by ref. 21 (10.16904/8).

The AVISO sea-surface height data by ref. 59 (https://www.aviso.altimetry.fr/en/data/products/auxiliary-products/mdt.html).

ERA5 monthly mean precipitation and evaporation rates by ref. 64 (https://cds.climate.copernicus.eu/cdsapp#!/search?type=dataset).

JRA55 monthly mean precipitation and evaporation rates by ref. 65 (https://rda.ucar.edu/).

OAFlux monthly mean evaporation rate by ref. 67 (https://oaflux.whoi.edu/).

GPCP monthly mean precipitation rate by ref. 66 (https://www.ncei.noaa.gov/).

GNIP yearly mean precipitation in Marion Island (https://www.iaea.org/services/networks/gnip).

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

## Acknowledgements
This project has received funding from the European Union's Horizon 2020 research and innovation program under grant agreement no. 821001, and from the European Union's Horizon Europe Funding Program for research and innovation under grant agreement Nr. 101060452. F.A.H. was supported by NSF's Southern Ocean Carbon and Climate Observations and Modeling (SOCCOM, PLR-1425989) Project, by NASA Grant 80NSSC19K1115, and by the European Union (ERC, VERTEXSO, 101041743). The authors gratefully thank the captain and the crew of the RV Marion Dufresne for their help in acquiring the long-term dataset as part of the OISO cruises and acknowledge INSU, IPEV and CNFH support for maintaining the OISO surveys. We also wish to acknowledge Catherine Pierre, Aïcha Naamar and Jérôme Demange for their skilled assistance in data analysis. The isotopic data were acquired at the L-CISE isotopic laboratory of OSU Ecce-Terra at LOCEAN, with support from OSU Ecce-Terra, LOCEAN and IPSL. This is a contribution to the LEFE Ker-Trend project.

## Author contributions
C.H.A., J.B.S. and G.R. directed the analysis of the datasets and share the responsibility for writing the manuscript with contributions from F.A.H., E.P., C.C.C., F.M., J.M. and C.S. C.L.M. and N.M. are the co-investigators of the ongoing OISO project, and produced the data synthesis with G.R. All authors contributed to the final version of the manuscript.

## Funding

## Competing interests
The authors declare no competing interests.
