## [Peer Review File · Nature Communications]

Isotopic evidence for an intensified hydrological cycle in the Indian sector of the Southern OceanReviewer #1 (Remarks to the Author):

The manuscript entitled "Isotopic evidence for an intensified hydrological cycle in the Indian sector of the Southern Ocean" assembled multidecadal sea surface salinity and oxygen isotope measurements, revealing contrasting trends in the SSS between subtropical and subpolar Southern Ocean in the Indian sector. Using a mixed layer freshwater budget along with the oxygen isotope measurements, authors decomposed the contribution to the mixed layer freshwater changes into P-E, glacial meltwater, and sea ice melt/formation where they found an increase in the net precipitation in the subpolar and increase in net evaporation in the subtropical region. I think the results presented in this manuscript is robust and of significance in understanding the freshwater changes in the Southern Ocean in the context of historical condition and eludes into the future projection under the changing climate. I have some minor comments and questions and I recommend acceptance of this manuscript for publication with them being addressed.

Minor Comments:

1. This work used a 1D framework to disentangle the mixed layer freshwater budget, one obvious caveat of this framework is to neglect the dynamic contribution associated with the changes in the circulation, specifically the meridional displacement of the ACC. Does author expect negligible net zonally/meridionally advective contribution to the mixed layer freshwater content? I imagine that the streamwise practice can to some degree mitigate the desire of quantifying the advection contribution. Nevertheless, this should be commented on somewhere in the manuscript (e.g., the Methods).
2. How does the author obtained the P-E trend ($37 \pm 5 \text{ mm yr}^{-1}$ per decade) for subpolar sector before computing glacial meltwater and sea ice, given that both P-E and glacial meltwater are unknown in the equation?
3. The vertical distribution of meltwater is neglected here when incorporating the glacial meltwater into the matrix. Author commented on the horizontal inhomogeneity of meltwater distribution, can author also comment on the fact that there is vertical distribution of meltwater? Even though it won't change the fact that this is still the upper bound estimate of meltwater contribution.
4. More of an outlook perspective: Author made comment that these results are not able to be extrapolated to other parts of the Southern Ocean. I can also imagine that the streamwise practice will collapse in region dominated by gyre regime. Given that the reanalyses P-E are highly uncertain, does this mean that there is little can be said about the pan-Antarctica freshwater contribution from glacial meltwater, sea ice and P-E on the observational basis?

Reviewer #2 (Remarks to the Author):

Akhoudas et al use observations of salinity and seawater oxygen isotope to reconstruct changes in the hydrological cycle in the Indian Sector of the Southern Ocean. Detecting changes in the hydrological cycle is challenging, especially in the Southern Ocean where observations are very sparse. The authors identify an intensification of the hydrological cycle in this region by combing unique datasets that include a long (since the 1990s) time series of seawater oxygen isotopes. The analysis is accurate and the results are very interesting and appropriate for Nature Communications. In particular, these findings highlight how seawater isotopes can provide information that other measurements can't, especially in terms of intensification of the hydrological cycle, changes in sea ice and ice sheets.

I have a main comment and some minor comments below.

- Line 229. The dataset used for sea ice freshwater fluxes covers the period 1993-2008. However, sea ice has shown major changes after 2008, in particular after 2016 with a strong decline. Could these changes impact salinity and O18 in your study region (e.g. through local or upstream changes)?

Minor Comments

- Figure 1d. Does the black line show the regression south of 46°S? Please specify in the caption.

- Line 101-126: I would include the equations in the main text, or explain a bit more how P-E is estimated.

- Line 167-169. It might be worth mention that seawater oxygen isotopes show a much stronger signal of glacial meltwater near the Antarctic coast (both in East and West Antarctica), corroborating your conclusion that the impact of glacial meltwater is (at present) mostly felt in the continental shelf region.

- Line 177: I would add here a reference and a brief discussion of what shown in Morrow and Kestenare (2014; Journal of Marine Systems) as they suggest that changes in surface salinity south of Australia are not only associated with precipitation.

- Line 185: Does the depth of the measurements influence the spatial pattern and time changes? I guess not much as the data are from the mixed layer, but better to include few words about this.

- Line 238: Changes in the input of the freshwater from the Antarctic Ice Sheet is not easy to extract from observations. It occurs mostly through ice shelf melting and iceberg calving. What about combining estimate of ice shelf basal melt (e.g. Adusumilli et al., 2020; Nat. Geosci.) and estimate iceberg discharge (Greene et al., 2022; Nature)?

- The impact of ice sheet melting is small this "far north" and therefore changes in S and O18 are likely due to changes in sea ice or P-E. These contributions are linked as snow accumulates on top of sea ice. Sea ice is advected to the north and it can carry snow into the study region and then melt. This is still part of the hydrological cycle. I would mention this, potentially discussing whether the signal you are observing might capture both local and non-local contributions.

Reviewer #3 (Remarks to the Author):

Review of "Isotopic evidence for an intensified hydrological cycle in 1 the Indian sector of the Southern Ocean" by Camille Hayatte Akhoudas et al.

General Comments: This paper tackles the evidence for an accelerated hydrological cycle using ocean salinity and oxygen isotopes. Speeding up of the global hydrologic cycle due to climate change is a clear outcome of the assessment of the scientific literature from IPCC. However the question of the role of the acceleration of the water cycle is quite unclear because of the confounding processes in the Antarctic (and Arctic region) from freshwater transport of sea-ice, changes in precipitation, and the potential contributions of the ice sheet melt adding to these processes.

The scientific literature is unclear on which processes are significant in the freshwater

balance around Antarctica and indeed that is the virtue of this paper. The wider research community is focussed on the contributions of the Antarctic mass loss (and its freshwater signal) to ocean salinity.

This observational paper has the virtue of using oxygen isotopes and ocean salinity change (with a simple) ocean surface water balance to make the case for the varying contributions of these various freshwater processes north and south of 46S. The oxygen isotopes have different fractionated states depending on the material, meteoric ice, sea-ice or rain. The surprising result to me from this paper is that in the region south of 46S, where there are all three processes operating, that the subpolar waters are freshening primarily from changes in P-E, with the second largest term being the reduction in the northward transport of sea-ice (and thus melt) and the even smaller contribution from glacial melt water. So the surprise (and also confirmation of the large-scale freshening of the surface waters) is that this freshening signal is mainly driven by increased precipitation (minus evaporation). And that the two other terms are weaker by an order of magnitude and coincidentally cancel each other out. I like this result, and just emphasises the importance of taking into account precipitation, and that the changes in sea ice and glacial meltwater are still weak contributions to the water cycle changes. These results provide a consistency check on the observed (but weak) circumpolar freshening of surface salinity.

So the innovation is the use of a unique surface data set that has this capacity to differentiate the different sources of freshwater sources and affirm that there is direct observation evidence in this region.

I have some doubts about the reliability of the overall estimate (see the specific comments). The trends in Figure 3, in the sense if I removed some of the values the trends could be much reduced or change sign, seem fragile and would benefit from a generalised cross validation.

The appropriateness of the salinity balance equations. Equations 4 and 5 are highly idealised versions of the salinity balance, tailored to make use of the relatively low amount of data in this time series. Is it possible to use the mean fields from a re-analysis product (eg ECCO) and include a tracer advection of oxygen isotope as a way to verify the surface salinity changes?

Overall I am very positive about this paper, and actually think it is publishable.

Specific Comments:

While I like the result for this sector, there are a number of significant questions to address in the overall analysis. I discuss them here.

Line 54: ... northward OF the ACC ...

Line 66: ... same processes ... The comment here is that the discussion upto this point has not recognised the contributions of advection to the surface processes. I know its coming because later in the paper transport of sea-ice is discussed. Surface advection of the surface waters needs to explicitly discussed at this stage and shown to be taken into account.

Line 76 ... a weighted linear regression ...

Line 91-93 I like the recognition of the salinity changes tend to be along streamlines. Streamlines of flow or dynamic height. Please be clear here.

Line 101: (and elsewhere). There should be an explicit statement about what P-E means, ie positive P-E is a freshwater flux into the ocean (say). The audience is more general.

Line 104: it is asserted that influence on sea-ice is limited to the southern region, but there is no evidence to support. The westerly winds and Ekman should carry these surface waters further northwards. Maybe a Mazloff et al paper, on ECCO could be cited as evidence, if appropriate. This is a key point for the argument that follows and the separation of the analysis of surface salinity.

Line 106 to 112: It was not clear to me how equations 4 and 5 are actually solved (and this comment includes my reading of the supplementary material). How was the uncertainty of the oxygen isotopes taken into account. Did the bootstrapping including working across the variation in the oxygen isotopes values (eg glacial melt -24±3 per mill.)

Line 130 to 132: The entire discussion here is about the surface salinity value. A salinity budget should of course include at least the thickness of the surface mixed layer, and if this had thinned at later times, then there would be even more fresh values. Has this thickness been taken into account or assessed explicitly?

Line 141-143: The error bars are carefully given for the rates of change per decade (either for oxygen isotopes or surface salinity). The error bars are often relative larger for the northern region compared with the southern region, and different for the three processes. Can some insight be given to variations in the relative accuracy for surface salinity and oxygen isotopes be given?

Line 149-151: Equations 4 and 5 depend on the mean isotopic values and the error ranges, are these ranges taken into account in the bootstrapping solutions in surface salinity error bars?

Line 159-160: “.. leading to a net surface ocean freshening...”, I think it needs to be made clear that this ocean freshening is for the region where the sea-ice is transported too, rather than say the coastal or polynya regions where the sea-ice is formed “surface salinity” would increase from this contribution.

Line 167: “... only a limited influence of an increase in Antarctic discharge on the surface water ...”

Line 176-176: This last sentence is at odds with the earlier assertion that the larger scale surface ocean salinity map and the changes described in this region is consistent, and here they should not be extrapolated to the whole southern ocean. And inconsistent with the discussion in the methods section. I think it is useful to provide the back of envelope estimates for the whole of Southern from this sector, keeping in mind these are trends that are derived from data nearly 30 years.

Couple of specific comments: the paper reports the average surface salinity changes. But actually what is changing is the depth integrated mixed layer salt content, and P-E is reported as g/kg per decade. So it would be helpful to have a table of the more common flux changes, so P-E would be mm/year per decade. This would allow a comparison with the NCEP or ERA P-E values. In this work I have not seen a table or the equivalent P-E values from NCEP (I note the discussion in methods) compared with the results from this work. The inhomogeneity and inconsistencies become a case that support these measurements. Such a table would combine the re-analyses products, the ice melt and the sea-ice transport values altogether.

There is no discussion of the trends of temperature in the measurements. Has the mixed layer thinned and so these freshening values in the southern ocean are a result of a thinner mixed layer. It's unclear whether the mixed layer depth is variable in space (in the salinity balance equations).

Figure 1. Last sentence in figure caption. I wondered whether the black line was being used.

Figure 3. Looking in these error bars I wondered how robust the trends are in the trends, and if one was to remove a few values from each panel, whether the estimates of the trends would be as stable as suggested by the linear trends with associated errors. There are relatively few degrees of freedom, and looking at the seasonal values, there is some serial correlation. It would be good to test the reliability of the trends here by subsetting the values to see if the estimates are stable, and then say so.

We would like to thank the reviewers for their thorough assessment of our manuscript. Their comments and suggestions have significantly improved the manuscript. Our direct responses to all reviewer comments and suggestions are below (in blue). Line numbers correspond to the new version of the manuscript, with all changes tracked.

Reviewer #1 (Remarks to the Author):

The manuscript entitled “Isotopic evidence for an intensified hydrological cycle in the Indian sector of the Southern Ocean” assembled multidecadal sea surface salinity and oxygen isotope measurements, revealing contrasting trends in the SSS between subtropical and subpolar Southern Ocean in the Indian sector. Using a mixed layer freshwater budget along with the oxygen isotope measurements, authors decomposed the contribution to the mixed layer freshwater changes into P-E, glacial meltwater, and sea ice melt/formation where they found an increase in the net precipitation in the subpolar and increase in net evaporation in the subtropical region. I think the results presented in this manuscript is robust and of significance in understanding the freshwater changes in the Southern Ocean in the context of historical condition and eludes into the future projection under the changing climate. I have some minor comments and questions and I recommend acceptance of this manuscript for publication with them being addressed.

We thank the reviewer for this careful reading of the manuscript and for providing suggestions to improve the manuscript. The comments from the reviewer have been addressed in the manuscript as detailed below.

Minor Comments:

1. This work used a 1D framework to disentangle the mixed layer freshwater budget, one obvious caveat of this framework is to neglect the dynamic contribution associated with the changes in the circulation, specifically the meridional displacement of the ACC. Does author expect negligible net zonally/meridionally advective contribution to the mixed layer freshwater content? I imagine that the streamwise practice can to some degree mitigate the desire of quantifying the advection contribution. Nevertheless, this should be commented on somewhere in the manuscript (e.g., the Methods).

We thank the reviewer for this constructive comment. We agree that a meridional shift of the fronts, could potentially affect our results. However, our results, in terms of long-term salinity and $\delta^{18}\text{O}$ trends, cannot be explained by the influence of meridional shifts of the fronts. First and foremost, while there has been an active debate of the ACC meridional position in response to a southward shift of the atmospheric jet, most recent works have led to the consensus that ACC fronts have not shifted southward in the past decades, as described in the review by Chapman et al. (2020) or in the latest IPCC assessments (Meredith et al., 2019; Fox-Kemper et al., 2021). Second, even if we made the assumption that the ACC has been shifted southward, the resulting local trend in salinity and $\delta^{18}\text{O}$ would be opposite to what we actually observed. Indeed, a southward shift of the ACC fronts would move saltier waters (and enriched in oxygen-18) from the north to the south. So, south of 46°S, this would produce increasing salinity and $\delta^{18}\text{O}$ trends, in stark contrast with what we observe. For these reasons, a meridional

movement of the ACC fronts cannot explain our results and we have now clarified this aspect in the ‘Methods’ section and in the ‘Linking changes of freshwater forcing to changes of surface ocean properties’ subsection with additional references - Line 233:

‘Here we use Eqs. 4 and 5 to determine long-term changes in P-E consistent with our best estimates of the other terms of the equations. In these equations, horizontal advection is neglected, as the coordinate system that we use follows the climatological mean position of the ACC fronts. Multidecadal changes in horizontal advection could be induced by, for example, a meridional displacement of the ACC fronts. However, recent assessments conclude that large meridional shifts of Southern Ocean fronts are unlikely over the past decades^{2,46,56}. Any small potential southward shift of fronts would in fact lead to a signal of opposite sign than what we observe, i.e., higher salinities towards the south, and are thus unlikely to affect the overall conclusion of our study. In addition, the possibility of a slight change in the fronts position or any other approximation made in these equations are, to some extent, taken into consideration by repeating 50 000 times the computation of Eqs. 4 and 5, but varying all terms within their uncertainties. We then retain all solutions that provide consistent estimates of F_{P-E} when computed by Eq. 4 compared to when computed by Eq. 5 (within a range of $\pm 5 \text{ mm yr}^{-1}$ per decade).’

And in the ‘Streamwise coordinate system’ subsection – Line 270:

‘In this study, climatological means, anomalies, and trends are computed within dynamically consistent regions with regards to the mean position of fronts, using a streamwise coordinate system. This streamwise coordinate system enables us to properly investigate ocean properties in this meandering region of the Southern Ocean, but neglects any potential meridional advection into the mixed layer. Because the geostrophic circulation follows contours of dynamic topography, we chose to work in bins of constant mean dynamic topography rather than bins of latitude⁵⁸.’

2. How does the author obtained the P-E trend ($37 \pm 5 \text{ mm yr}^{-1}$ per decade) for subpolar sector before computing glacial meltwater and sea ice, given that both P-E and glacial meltwater are unknown in the equation?

We agree with the reviewer that this part was not clear in the manuscript. We did not obtain the P-E trend for the subpolar sector before including glacial meltwater and sea ice changes in our calculation. We first compute changes of all forcings, i.e., P-E, glacial melt and sea ice. The latter two are first constrained by the literature estimates, and then we select all consistent solutions across Eqs. 4 and 5 that explain the observed salinity and $\delta^{18}\text{O}$ trends in the subpolar sector. This has now been clarified in the manuscript - Line 139:

‘When allowing for all forcing perturbations, i.e., P-E, glacial melt and sea ice, and selecting all consistent solutions across Eqs. 4 and 5, we find that the best estimate of P-E changes is an increase of $33 \pm 11 \text{ mm yr}^{-1}$ per decade, with a mean precipitation $\delta^{18}\text{O}$ of $-11 \pm 3\%$. In this solution, glacial melt changes by $11 \pm 4 \text{ mm yr}^{-1}$ per decade, with a mean ice sheet $\delta^{18}\text{O}$ of $-25 \pm 3\%$, and sea ice melt decreases by $-12 \pm 2 \text{ mm yr}^{-1}$ per decade.’

3. The vertical distribution of meltwater is neglected here when incorporating the glacial meltwater into the matrix. Author commented on the horizontal inhomogeneity of meltwater

distribution, can author also comment on the fact that there is vertical distribution of meltwater? Even though it won't change the fact that this is still the upper bound estimate of meltwater contribution.

We thank the reviewer for raising this point. Indeed, vertical mixing of the anomalies is expected, especially during convection in the upper ocean in winter. This process is likely to distribute the anomalies over a wide vertical range, and is key to structuring the impact of the freshwater changes on the upper layer. However, the hypothesis here is that a significant amount of meltwater will accumulate and remain in the upper ocean mixed layer. We have made this point clearer in the manuscript - Line 317:

‘Assuming that this glacial meltwater is evenly spread within the mixed layer over the entire subpolar region (south of the mean dynamic topography contour of -0.2 m; corresponding to *lat* 46°S; area of $\sim 3.5 \cdot 10^7$ km²), it converts into a freshwater flux trend ranging between 5 and 17 mm yr⁻¹ per decade from 1993 to 2021. However, we note that those changes in glacial meltwater flux are likely not evenly spread within the mixed layer over the entire subpolar region, and most likely have an impact to a greater extent in the coastal west Antarctic region than in the northern edge of the subpolar region of the Indian sector of the Southern Ocean.’

4. More of an outlook perspective: Author made comment that these results are not able to be extrapolated to other parts of the Southern Ocean. I can also imagine that the streamwise practice will collapse in region dominated by gyre regime. Given that the reanalyses P-E are highly uncertain, does this mean that there is little can be said about the pan-Antarctica freshwater contribution from glacial meltwater, sea ice and P-E on the observational basis?

We agree with the reviewer that an alternative approach would need to be developed depending on the dynamics of the region of interest. In a gyre regime, one could however assume that properties are likely more homogeneous and use that as a strength to analyze samples spread over the gyre region. It is difficult for us to comment on an approach that would need to be refined based on the specificity of the case. We note that the main limitation, as far as we can tell, is the lack of long-term datasets with consistent spatial and temporal coverage.

Reviewer #2 (Remarks to the Author):

Akhoudas et al use observations of salinity and seawater oxygen isotope to reconstruct changes in the hydrological cycle in the Indian Sector of the Southern Ocean. Detecting changes in the hydrological cycle is challenging, especially in the Southern Ocean where observations are very sparse. The authors identify an intensification of the hydrological cycle in this region by combining unique datasets that include a long (since the 1990s) time series of seawater oxygen isotopes. The analysis is accurate and the results are very interesting and appropriate for Nature Communications. In particular, these findings highlight how seawater isotopes can provide information that other measurements can't, especially in terms of intensification of the hydrological cycle, changes in sea ice and ice sheets.

We thank the reviewer for this careful reading of the manuscript and for providing suggestions to improve the manuscript. The comments from the reviewer have been addressed in the manuscript as detailed below.

I have a main comment and some minor comments below.

- Line 229. The dataset used for sea ice freshwater fluxes covers the period 1993-2008. However, sea ice has shown major changes after 2008, in particular after 2016 with a strong decline. Could these changes impact salinity and O18 in your study region (e.g. through local or upstream changes)?

We thank the reviewer for raising this point. Indeed, sea ice in the Southern Ocean experienced an unprecedented retreat in austral spring 2016 (Turner et al., 2017). Furthermore, it seems that there is a temporal lag between signals imparted in the sea ice formation zone in the south and the impacted area further north. For instance, Meredith et al. (2023) show patterns of salinity and $\delta^{18}\text{O}$ anomalies in the northern Weddell Sea in response to anomalous sea ice cover in recent years. They found that the maximal oceanic imprint from sea ice in the Weddell Sea occurs 1-2 years after the rapid decline in 2016. Similarly, in our region of interest, a signal imparted further south in the sea ice zone would likely take some years to reach the northern edge of the subpolar sector where our salinity and $\delta^{18}\text{O}$ data were collected. If that was the case, we should observe a positive salinity anomaly without much signal on $\delta^{18}\text{O}$ around 2016-2018 in the southern sector. We indeed observe a large positive salinity anomaly in 2017 in Figure 3c, which might be consistent with the large sea ice anomaly of 2016. But the attribution remains speculative. With more observation in the future, we will be able to fully address this question and seek a potential regime shift in salinity and $\delta^{18}\text{O}$. But we believe that would require additional data and a dedicated study. What could be questioned is how such change would affect the range of the sea ice trend used to constrain our equations 4 and 5. As noted by the reviewer, the sea ice product we used to derive this range covers the period 1993-2008. However, as shown in Fogt et al., 2022 (see their Figure 3k), because the anomalous decline is limited to only a few years at the end of our time series, the sea ice trend computed over the entire extent of the satellite time period is not much affected by ending the trend computation before or after 2016. In addition, we use a relatively loose range for sea ice freshwater flux with an uncertainty of 33 % (namely $-12 \pm 4 \text{ mm yr}^{-1}$ per decade), which should allow for a slight change in the direction of the flux at the end of the time series.

We added a few words in the discussion referring to the sea ice decline from 2016. Line 183:

‘This feature is consistent with estimates from Ref.²¹ for this region, showing a decline in the northward sea ice transport in the Indian sector of the Southern Ocean and upstream of this region. The ocean’s salinity response to this signal in a model⁵ is a clear salinification, consistent with the slight negative salinity trend accompanied by the strong negative $\delta^{18}\text{O}$ trend south of *lat* 52°S. Despite large regional variations, the total Antarctic sea ice cover slightly increased since the late 1970s⁴⁷ and sharply decreased in 2016⁴⁸, likely affecting the salinity anomalies towards the end of our time series (Fig. 3c). This sea ice decline, which continued after 2016, might become more prominent in the future, but should be covered by the relatively loose range of the estimate and its associated error of 33 % applied in this study.’

Minor Comments

- Figure 1d. Does the black line show the regression south of 46°S? Please specify in the caption.

We removed the black line and the corresponding caption as this was not a relevant information for our study.

- Line 101-126: I would include the equations in the main text, or explain a bit more how P-E is estimated.

This is a good point. We prefer to not include all the equations in the main text, but we have now included the simplified version of them for the subtropical sector, to better guide the reader, and point them to the methods section for the full derivation of the equations. We have clarified the text to better explain how the equations are computed and how P-E is estimated. Line 110:

‘Qualitatively, long-term salinification in the subtropics can be explained by a decrease in P-E, i.e., a decline of freshwater flux into the ocean. In contrast, long-term freshening in the subpolar sector could be explained by an increasing P-E, i.e., an amplification of freshwater flux into the ocean, a shift in sea ice regime, an increasing rate of ice sheet mass loss, or any combination of these. The weak correlation between salinity and $\delta^{18}\text{O}$ in the southern region, and the strong correlation in the subtropics (Fig. 2d) suggest that the influence of sea ice (which tends to disrupt the correlation between salinity and $\delta^{18}\text{O}$) is limited to the subpolar region and does not considerably influence the subtropical region. The surface water influenced by sea ice in the subpolar sector is actually advected northward through meridional Ekman transport, but most of it subducts into intermediate and mode waters along the ACC fronts³⁸, and does not remain in the surface layer in the subtropical sector. Consequently, we aim to explain the observed changes in surface salinity and $\delta^{18}\text{O}$ by a combination of changes in P-E, sea ice, and glacial meltwater input in the subpolar sector, and solely by changes in P-E in the subtropical sector (see Eqs. 4 and 5 in Methods). To do so, in the subtropical sector, we compute changes in P-E ($\Delta F_{\text{P-E}}$) from observed changes in salinity (ΔS) and $\delta^{18}\text{O}$ ($\Delta\delta^{18}\text{O}$):

$$\Delta F_{\text{P-E}} = h\Delta S / (S_{\text{P-E}} - S_0),$$
$$\Delta F_{\text{P-E}} = h\Delta\delta^{18}\text{O} / (\delta^{18}\text{O}_{\text{P-E}} - \delta^{18}\text{O}_0).$$

where h is the depth of the mixed layer, $S_{\text{P-E}}$ and $\delta^{18}\text{O}_{\text{P-E}}$ are the salinity and $\delta^{18}\text{O}$ of precipitation, S_0 and $\delta^{18}\text{O}_0$ the mean mixed layer salinity and $\delta^{18}\text{O}$. We repeat the computation of each of these two equations 50 000 times, but at each iteration we randomly perturb estimates of ΔS , $\Delta\delta^{18}\text{O}$, $S_{\text{P-E}}$, $\delta^{18}\text{O}_{\text{P-E}}$, within their error ranges (see Table 1). We therefore obtain a set of 50 000 estimates of $\Delta F_{\text{P-E}}$ consistent with observed changes in ΔS , and another set of 50 000 estimates of $\Delta F_{\text{P-E}}$ consistent with observed changes in $\Delta\delta^{18}\text{O}_{\text{P-E}}$. We then select all $\Delta F_{\text{P-E}}$ estimates that are consistent across the two set of solutions (within a range of $\pm 5 \text{ mm yr}^{-1}$ per decade). Our best estimate and associated error of $\Delta F_{\text{P-E}}$ is the mean \pm standard deviation of these sub selected solutions. We also compute the corresponding best estimate and error of $\delta^{18}\text{O}_{\text{P-E}}$ with the same strategy (see Table 2). We find that our best estimate of changes in P-E resulting from this bootstrap exercise is a reduction of $-71 \pm 47 \text{ mm yr}^{-1}$ per decade for

precipitation with a mean isotopic composition of $-7 \pm 2\text{‰}$. We note that the impact of multidecadal changes in mixed layer depth is neglected in our calculation. Ref.¹² showed that in this region the mixed layer depth has deepened at a rate of $2 \pm 4 \%$ per decade, which is a relative change that would be more than 3 times lower than the relative change of P-E, based on atmospheric reanalysis estimates (ERA5, JRA55 and GPCP/OAFlux).

In the subpolar sector, we repeat the exact same strategy, except that we have additional terms associated to the possible influence of sea ice and ice sheet (see Eqs. 4 and 5). The bootstrap exercise is performed assuming a range for isotopic signatures of the Antarctic glacial meltwater^{29,39} and sea ice⁴⁰ (see Table 1). We start by searching for solutions by assuming no changes have occurred in P-E, to test the hypothesis that observed changes could be solely driven by sea ice regime and glacial meltwater. No consistent solutions are found across Eqs. 4 and 5 under this assumption; so, we conclude that changes in P-E must be involved in the observed surface ocean trends. When allowing for all forcing perturbations, i.e., P-E, ice sheet and sea ice, and selecting all consistent solutions across Eqs. 4 and 5, we find that the best estimate of P-E changes is an increase of $33 \pm 11 \text{ mm yr}^{-1}$ per decade with a mean precipitation $\delta^{18}\text{O}$ of $-11 \pm 3\text{‰}$. In this solution, glacial melt changes by $11 \pm 4 \text{ mm yr}^{-1}$ per decade, with a mean ice sheet $\delta^{18}\text{O}$ of $-25 \pm 3\text{‰}$, and sea ice melt decreases by $-12 \pm 2 \text{ mm yr}^{-1}$ per decade.’

- Line 167-169. It might be worth mention that seawater oxygen isotopes show a much stronger signal of glacial meltwater near the Antarctic coast (both in East and West Antarctica), corroborating your conclusion that the impact of glacial meltwater is (at present) mostly felt in the continental shelf region.

We thank the reviewer for this comment. We added a reference to the stronger signal of glacial meltwater near the Antarctic coast with the corresponding literature, i.e., Jacobs et al., 2002; Akhondas et al., 2020; 2021. Line 197:

‘Second, our findings suggest only a limited influence of an increase in Antarctic ice discharge on the surface water properties in the subpolar ocean, away from the continent. This is corroborated by the observational evidence of the depleted $\delta^{18}\text{O}$ signal near the Antarctic coast^{39,49}, the increased glacial meltwater mostly affecting the waters along the continental shelf. The weak response to the open ocean to this coastal imprint is likely related to the significant mixing with higher $\delta^{18}\text{O}$ waters across the shelf break where the depleted $\delta^{18}\text{O}$ signal of waters flowing northward rapidly decreases⁵⁰.’

- Line 177: I would add here a reference and a brief discussion of what shown in Morrow and Kestenare (2014; Journal of Marine Systems) as they suggest that changes in surface salinity south of Australia are not only associated with precipitation.

We thank the reviewer for this insightful comment. We added a brief discussion including the interesting Morrow and Kestenare (2014) paper at the end of the discussion section. Line 210:

‘They should not be extrapolated to the wider Southern Ocean, since changes in surface freshwater fluxes and their imprint on the ocean can have a spatially complex pattern, such as e.g., identified for sea ice changes⁵, and, as an example, the region south of Tasmania where local surface salinity changes might not be driven by net precipitation variations, as shown by Ref.⁵⁴. Interannual variability of surface salinity driven by strong events such as southward subtropical water flow might hide the contribution of local P-E forcing. The different responses

of local regime to direct or remote forcing are thus important to take into account in order to investigate surface changes in the Southern Ocean as a whole.'

• Line 185: Does the depth of the measurements influence the spatial pattern and time changes? I guess not much as the data are from the mixed layer, but better to include few words about this.

We thank the reviewer for this pertinent question. The analysis includes all data between 0- and 50-meters depth but in reality, there is not much data below 25- meters depth (i.e., about 20 such observations in the entire dataset). The zonally-averaged meridional profiles of salinity and $\delta^{18}\text{O}$ between 0- and 25-meters (green dots) and between 25- and 50-meters (blue dots) lie within the standard errors envelop (red dashed lines) of the zonally-averaged meridional profiles of salinity and $\delta^{18}\text{O}$ between 0- and 50-meters (see Figures below). This gives us confidence that the depth of the measurements does not influence the spatial pattern and time changes estimated from our analyses.

Figure - Meridional profiles of zonally averaged salinity (left) and $\delta^{18}\text{O}$ (right) and their standard errors in red dashed lines. Green dots are observations between 0- and 25-meters depth and blue dots are observations between 25- and 50-meters depth. Calculations are computed using an irregular mean dynamic topography grid with bin size ranging from 0.1 to 0.3 m.

We added a few words about this and the figure in the Supplementary Information. Line 245:

'In this study, we use surface observations of oxygen-18 isotopic composition ($\delta^{18}\text{O}$) and salinity (expressed as absolute salinity following TEOS-10), from underway sampling at a depth of about 5–7 m from the sea surface and from stations sampling between a depth of 0

and 50 m, in the Indian sector of the Southern Ocean (40°E–90°E and 30°S–60°S, Fig. 1a). These observations are from oceanographic research cruises undertaken during austral summer (December to February) from 1993 to 2021 (Fig. 1b). We note here that the depth of the measurements within the upper 50 meters does not influence the mean salinity and $\delta^{18}\text{O}$ surface fields (Supplementary Fig. 8).’

- Line 238: Changes in the input of the freshwater from the Antarctic Ice Sheet is not easy to extract from observations. It occurs mostly through ice shelf melting and iceberg calving. What about combining estimate of ice shelf basal melt (e.g. Adusumilli et al., 2020; Nat. Geosci.) and estimate iceberg discharge (Greene et al., 2022; Nature)?

We thank the reviewer for this helpful suggestion. We modified our glacial meltwater estimate according to Greene et al., 2022. As described in the ‘Ice sheet’ subsection of the ‘Methods’ section, we use the assessment of the latest IPCC report to estimate glacial meltwater increasing input from ice discharge at the grounding lines and from ice shelf thinning; $113 \pm 42 \text{ Gr yr}^{-1}$ per decade and 20 to 200 Gt yr^{-1} per decade, respectively, as the best estimates of plausible changes from 1993 and 2021. Greene et al., 2022 showed that Antarctica has experienced a net mass loss of $5874 \pm 396 \text{ Gt}$ of ice associated with ice front retreat between 1997 and 2021, approximately equal to ice shelf thinning. We then use a wide range of the total Antarctic glacial meltwater input including ice discharge at the grounding lines and doubled ice shelf thinning estimates in order to envelop the large uncertainties of these estimations, i.e., 200 to 600 Gt yr^{-1} per decade. Line 314:

‘In addition to these two components, ice mass loss associated with ice front retreat has been recently estimated as approximately equal to net ice mass loss due to ice shelf thinning⁶¹. Our best estimate of changes in total (grounding line plus double ice shelf thinning) Antarctic glacial meltwater input therefore ranges from about 150 to 550 Gt yr^{-1} per decade from 1993 to 2021, consistent with the estimate discussed in Ref.⁶². Assuming that this glacial meltwater is evenly spread within the mixed layer over the entire subpolar region (south of the mean dynamic topography contour of -0.2 m; corresponding to *lat* 46°S; area of $\sim 3.5 \cdot 10^7 \text{ km}^2$), it converts into a freshwater flux trend ranging between 5 and 17 mm yr^{-1} per decade from 1993 to 2021.’

- The impact of ice sheet melting is small this “far north” and therefore changes in S and O18 are likely due to changes in sea ice or P-E. These contributions are linked as snow accumulates on top of sea ice. Sea ice is advected to the north and it can carry snow into the study region and then melt. This is still part of the hydrological cycle. I would mention this, potentially discussing whether the signal you are observing might capture both local and non-local contributions.

We thank the reviewer for this pertinent comment. The freshwater signal we observe, captures both local and non-local contributions as by definition signal from the coast and the sea ice formation zone is advected to the north and so in the subpolar region of our study. The observed P-E signal in the subpolar region can also reflect non-local signal from the coast. Snow

accumulated on top of sea ice is indeed also part of the hydrological cycle as mentioned by the reviewer, contributing to the P-E signal that is advected to the north. From our study, it is thus impossible to disentangle the local contribution from the non-local contribution. We discuss this in the main text. Line 191:

‘Our study quantifies the contribution of changes in freshwater forcing that influence surface water properties in the Southern Ocean directly from observations. Here we analyze the ocean response to both local and non-local freshwater forcing perturbations as signal from sea ice and meteoric waters is advected northward. Furthermore, the process of northward freshwater transport by sea ice contribute also to non-local contribution of meteoric waters through snow accumulated on top of sea ice.’

Reviewer #3 (Remarks to the Author):

Review of “Isotopic evidence for an intensified hydrological cycle in the Indian sector of the Southern Ocean” by Camille Hayatte Akhoudas et al.

General Comments: This paper tackles the evidence for an accelerated hydrological cycle using ocean salinity and oxygen isotopes. Speeding up of the global hydrologic cycle due to climate change is a clear outcome of the assessment of the scientific literature from IPCC. However the question of the role of the acceleration of the water cycle is quite unclear because of the confounding processes in the Antarctic (and Arctic region) from freshwater transport of sea-ice, changes in precipitation, and the potential contributions of the ice sheet melt adding to these processes.

The scientific literature is unclear on which processes are significant in the freshwater balance around Antarctica and indeed that is the virtue of this paper. The wider research community is focussed on the contributions of the Antarctic mass loss (and its freshwater signal) to ocean salinity.

This observational paper has the virtue of using oxygen isotopes and ocean salinity change (with a simple) ocean surface water balance to make the case for the varying contributions of these various freshwater processes north and south of 46S. The oxygen isotopes have different fractionated states depending on the material, meteoric ice, sea-ice or rain. The surprising result to me from this paper is that in the region south of 46S, where there are all three processes operating, that the subpolar waters are freshening primarily from changes in P-E, with the second largest term being the reduction in the northward transport of sea-ice (and thus melt) and the even smaller contribution from glacial melt water. So the surprise (and also confirmation of the large-scale freshening of the surface waters) is that this freshening signal is mainly driven by increased precipitation (minus evaporation). And that the two other terms are weaker by an order of magnitude and coincidentally cancel each other out. I like this result, and just emphasises the importance of taking into account precipitation, and that the changes in sea ice and glacial meltwater are still weak contributions to the water cycle changes. These results provide a consistency check on the observed (but weak) circumpolar freshening of surface salinity.

So the innovation is the use of a unique surface data set that has this capacity to differentiate the different sources of freshwater sources and affirm that there is direct observation evidence in this region.

We thank the reviewer for this careful reading of the manuscript and for providing suggestions to improve the manuscript. The comments from the reviewer have been addressed in the manuscript as detailed below.

I have some doubts about the reliability of the overall estimate (see the specific comments). The trends in Figure 3, in the sense if I removed some of the values the trends could be much reduced or change sign, seem fragile and would benefit from a generalised cross validation.

We understand the concern of the reviewer regarding the robustness of the estimated trends based on a small amount of data. It is however an important point of our study to address sparsity and outlier issues by our approach that includes streamline coordinates and adequate uncertainty ranges. We focus on a time series of change within two wide regions of comparatively large data availability in the Southern Ocean as the first step to investigate surface property trends. By identifying larger regions of surface water masses, we omit some of the issues that might arise when investigating local-scale patterns, and which gives us robust trend estimates. The uncertainty associated with those estimated trends in the initial version were indeed based on the standard error of the fits. In order to reinforce the overall estimate, we apply bootstrapping to the trend calculations as suggested by the reviewer, taking 80 % of the values randomly 50 000 times as a generalized cross validation. The uncertainty is now computed as the standard deviation of the trends estimated with the bootstrapping. We added a few words about this approach in the ‘Methods’ section and replaced the figure 3 and figure 4 and the associated values in the main text. Line 281:

‘Trends are estimated by fitting linear regression models with their associated standard errors. The confidence level is established by computing the standard error of the regression. For trends performed in streamwise coordinate bins, we indicate trend values that are (i) less than half the computed standard errors, (ii) between half and one standard errors, (iii) between one and two standard errors, and (iv) more than two standard errors, to estimate the trend robustness with regards to the standard error. For trends produced in large sectors (north and south of *lat* 46°S), (i) we first compute local anomalies from the zonal-mean meridional profile by removing the mean of all observations in the streamwise coordinate bins; (ii) we then compute the annual 25, 50, 75 percentiles of anomalies in the large sector; (iii) finally, we fit a weighted linear regression model to the annual medians, weighted by $w = 1/IQR^2$, where IQR is the interquartile range (percentile 75 minus percentile 25), which provides an estimate of the long-term trend. Its associated uncertainty is then estimated as the standard error of the trend calculated using a bootstrap approach (repeated 50 000 times taking 80 % of the values randomly).’

The appropriateness of the salinity balance equations. Equations 4 and 5 are highly idealised versions of the salinity balance, tailored to make use of the relatively low amount of data in this time series. Is it possible to use the mean fields from a re-analysis product (eg ECCO) and include a tracer advection of oxygen isotope as a way to verify the surface salinity changes?

This is an excellent suggestion from the reviewer. Applying this approach would, however, require a significant amount of additional work, with its own caveats and strengths, and it is our opinion that it would be more suitable for a dedicated study.

Overall I am very positive about this paper, and actually think it is publishable.

We thank the reviewer for this positive comment.

Specific Comments:

While I like the result for this sector, there are a number of significant questions to address in the overall analysis. I discuss them here.

Line 54: ... northward OF the ACC ...

We thank the reviewer; we changed this part on line 56:

‘The observed surface properties abruptly change across *lat* 46° S, identified as the northern boundary of the Subantarctic Front (SAF-N)³³ and associated with the ACC. North of the SAF-N (north of *lat* 46°S), the two properties actually follow a clear linear slope depending on the evaporation and precipitation characteristics of the region³⁴ (Fig. 1d).’

Line 66: ... same processes ... The comment here is that the discussion upto this point has not recognised the contributions of advection to the surface processes. I know its coming because later in the paper transport of sea-ice is discussed. Surface advection of the surface waters needs to explicitly discussed at this stage and shown to be taken into account.

We agree with this comment. We added some words on the contribution of advection, which transport salinity and $\delta^{18}\text{O}$ characteristics set by surface fluxes. Line 68:

‘This high correlation is consistent with the observation of co-varying mean field (Fig. 1d), as a result of the two fields being mainly controlled by the same processes, evaporation and precipitation, along a fixed mixing line i.e., the amount of salinity changes for a given amount of $\delta^{18}\text{O}$ changes, both set by evaporation and precipitation characteristics³⁵. Salinity and $\delta^{18}\text{O}$ of a water parcel are indeed dependent on evaporation and precipitation (and possibly other sources of meteoric waters and freshwater flux from sea ice) that the water parcel encountered during its lifetime, which tends to be advected by ocean currents. Conversely to the northern region, but still consistent with the mean fields, both parameters exhibit a substantially weaker relationship, inferred from low linear correlation coefficients, in the region south of *lat* 46°S (Fig. 2d). This poor correlation between anomalies of salinity and $\delta^{18}\text{O}$ indicates that in addition to the local and non-local evaporation and precipitation, other processes are controlling surface salinity and $\delta^{18}\text{O}$. For example, while evaporation and precipitation have an impact on both parameters, $\delta^{18}\text{O}$ is only marginally affected by sea ice formation and melt with a negligible fractionation factor on the order of 2‰^{36,37}. Changes in sea ice formation and melt mostly impact the surface salinity. This feature of concomitant high salinity and low $\delta^{18}\text{O}$

variations shows the influence of sea ice fluxes in surface waters advected from further south. The subpolar sector south of *lat* 46°S is thus imprinted by local evaporation and precipitation processes as well as non-local sea ice and meteoric (net precipitation and glacial meltwater) fluxes.’

Line 76 ... a weighted linear regression ...

We thank the reviewer and this was corrected on line 85:

‘By fitting a weighted linear regression model [...].’

Line 91-93 I like the recognition of the salinity changes tend to be along streamlines. Streamlines of flow or dynamic height. Please be clear here.

We thank the reviewer, and this was corrected on line 100:

‘When looking at the spatial pattern from an independent analysis based on this much larger dataset, salinity changes are found to be mostly consistent along streamlines of dynamic height, which comforts our approach to investigate changes in streamwise coordinates (Supplementary Fig. 2).’

Line 101: (and elsewhere). There should be an explicit statement about what P-E means, ie positive P-E is a freshwater flux into the ocean (say). The audience is more general.

We agree with this comment. We added what P-E means in the first paragraph of the results section and we were more explicit elsewhere, when necessary, where P-E is stated. Lines 105 and 110:

‘However, there are inconsistencies among those products with a wide spread in precipitation minus evaporation (P-E)—positive P-E is a flux into the ocean—trends over the Southern Ocean (Supplementary Note 2 and Supplementary Fig. 5,6), which would introduce uncertainties when interpreting surface salinity changes observed in the Indian sector of the Southern Ocean.’

‘Qualitatively, long-term salinification in the subtropics can be explained by a decrease in P-E, i.e., a decline of freshwater flux into the ocean. In contrast, long-term freshening in the subpolar sector could be explained by an increasing P-E, i.e., an amplification of freshwater flux into the ocean, a shift in sea ice regime, an increasing rate of ice sheet mass loss, or any combination of these.’

Line 104: it is asserted that influence on sea-ice is limited to the southern region, but there is no evidence to support. The westerly winds and Ekman should carry these surface waters further northwards. Maybe a Mazloff et al paper, on ECCO could be cited as evidence, if appropriate. This is a key point for the argument that follows and the separation of the analysis of surface salinity.

The reviewer is correct that southern waters are carried northward by Ekman process, but as they are advected northward, they sink within intermediate and mode waters, as part of the upper overturning cell. The signal of sea ice and glacial meltwater is therefore spread at depth and not in the surface layer of the subtropical sector. The location of sinking is within the ACC

fronts, which corresponds to the sector near *lat* 46°S. This is a process that has been described from observations (e.g., Speer et al., 2018), forced model, or assimilated models like those produced by M. Mazloff and colleagues (see e.g., Cerovecki et al., 2013). We made the text clearer and added references in line 113:

‘The weak correlation between salinity and $\delta^{18}\text{O}$ in the southern region, and the strong correlation in the subtropics (Fig. 2d) suggest that the influence of sea ice (which tends to disrupt the correlation between salinity and $\delta^{18}\text{O}$) is limited to the subpolar region and does not considerably influence the subtropical region. The surface water influenced by sea ice in the subpolar sector is actually advected northward through meridional Ekman transport, but most of it subducts into intermediate and mode waters along the ACC fronts³⁸, and does not remain in the surface layer in the subtropical sector.’

Line 106 to 112: It was not clear to me how equations 4 and 5 are actually solved (and this comment includes my reading of the supplementary material). How was the uncertainty of the oxygen isotopes taken into account. Did the bootstrapping including working across the variation in the oxygen isotopes values (eg glacial melt -24+-3 per mill.)

We agree with the reviewer and we clarified in the main text how the equations are solved. The variation in the oxygen isotopes values is included in the bootstrap. We use wide ranges for $\delta^{18}\text{O}$ ice sheet corresponding to $-30 \pm 10\%$, for $\delta^{18}\text{O}$ precipitation in the subpolar region of $-10 \pm 5\%$ and for $\delta^{18}\text{O}$ precipitation in the subtropics of $5 \pm 5\%$, based on literature. These values are shown in Table 1 and mentioned in the ‘Results’ and ‘Methods’ sections. We have revised the text and hope this will clarify our methodology and satisfy the reviewer. Line 118 and 134:

‘Consequently, we aim to explain the observed changes in surface salinity and $\delta^{18}\text{O}$ by a combination of changes in P-E, sea ice, and glacial meltwater input in the subpolar sector, and solely by changes in P-E in the subtropical sector (see Eqs. 4 and 5 in Methods). To do so, in the subtropical sector, we compute changes in P-E ($\Delta F_{\text{P-E}}$) from observed changes in salinity (ΔS) and $\delta^{18}\text{O}$ ($\Delta\delta^{18}\text{O}$):

$$\Delta F_{\text{P-E}} = h\Delta S / (S_{\text{P-E}} - S_0),$$

$$\Delta F_{\text{P-E}} = h\Delta\delta^{18}\text{O} / (\delta^{18}\text{O}_{\text{P-E}} - \delta^{18}\text{O}_0).$$

where h is the depth of the mixed layer, $S_{\text{P-E}}$ and $\delta^{18}\text{O}_{\text{P-E}}$ are the salinity and $\delta^{18}\text{O}$ of precipitation, S_0 and $\delta^{18}\text{O}_0$ the mean mixed layer salinity and $\delta^{18}\text{O}$. We repeat the computation of each of these two equations 50 000 times, but at each iteration we randomly perturb estimates of ΔS , $\Delta\delta^{18}\text{O}$, $S_{\text{P-E}}$, $\delta^{18}\text{O}_{\text{P-E}}$, within their error ranges (see Table 1). We therefore obtain a set of 50 000 estimates of $\Delta F_{\text{P-E}}$ consistent with observed changes in ΔS , and another set of 50 000 estimates of $\Delta F_{\text{P-E}}$ consistent with observed changes in $\Delta\delta^{18}\text{O}_{\text{P-E}}$. We then select all $\Delta F_{\text{P-E}}$ estimates that are consistent across the two set of solutions (within a range of $\pm 5 \text{ mm yr}^{-1}$ per decade). Our best estimate and associated error of $\Delta F_{\text{P-E}}$ is the mean \pm standard deviation of these sub selected solutions. We also compute the corresponding best estimate and error of $\delta^{18}\text{O}_{\text{P-E}}$ with the same strategy (see Table 2).’

‘In the subpolar sector, we repeat the exact same strategy, except that we have additional terms associated to the possible influence of sea ice and ice sheet (see Eqs. 4 and 5). The bootstrap

exercise is performed assuming a range for isotopic signatures of the Antarctic glacial meltwater^{29,39} and sea ice⁴⁰ (see Table 1)'.

Line 130 to 132: The entire discussion here is about the surface salinity value. A salinity budget should of course include at least the thickness of the surface mixed layer, and if this had thinned at later times, then there would be even more fresh values. Has this thickness been taken into account or assessed explicitly?

We agree with the reviewer that the depth of the mixed layer must be taken into account because surface flux is diluted over the mixed layer. We actually do consider this effect (see for instance Equation 1-5 where h is the mixed layer depth). We do, however, neglect the impact of long-term change in the thickness as a compromise to reduce the complexity of the problem, and because we have reasons to believe that changes in the mixed layer depth is small compared to change in the flux. Indeed, Sallée et al., 2021 showed that in this region the mixed layer depth has changed (deepened) at a rate of a few meters per decade: their estimated mean trend south of lat 46°S in our region of interest, between 30-60°S and 40-90°E, is 3 meters per decade from 1970 to 2018. Such rate of change corresponds to only 2 ± 4 % per decade, when compared to the climatological mean depth. This is a relative change of more than 3 times lower than the relative changes of P-E, based on atmospheric reanalysis estimates (ERA5, JRA55 and GPCP/OAFlux). In addition, Sallée et al., 2021 described a deepening of the mixed layer in our region of interest which would lead to a dilution of the freshwater signal. So, if anything, the impact of the change in mixed layer thickness should underestimate the role of fluxes, and could not explain the observed salinity and $\delta^{18}O$ changes without invoking changes in the fluxes (e.g., more freshwater inputs would be needed to explain the same change in surface salinity and $\delta^{18}O$, because it would be increasingly more diluted over time). Line 130:

'We note that the impact of multidecadal change in mixed layer depth is neglected in our calculation. Ref.¹² showed that in this region the mixed layer depth has deepened at a rate of 2 ± 4 % per decade, which is a relative change that would be more than 3 times lower than the relative change of P-E, based on atmospheric reanalysis estimates (ERA5, JRA55 and GPCP/OAFlux).'

Line 141-143: The error bars are carefully given for the rates of change per decade (either for oxygen isotopes or surface salinity). The error bars are often relative larger for the northern region compared with the southern region, and different for the three processes. Can some insight be given to variations in the relative accuracy for surface salinity and oxygen isotopes be given?

We thank the reviewer for this comment. As we can see on the salinity- $\delta^{18}O$ diagram (Figure 1d in the main text) and the meridional profiles of zonally averaged salinity and $\delta^{18}O$ (Figure 2a and b), the ranges of salinity and $\delta^{18}O$ are wider north of lat 46°S than south of it with 35 ± 0.34 g kg⁻¹ and 0.34 ± 0.18 ‰ between 30°S and 46°S and, 33.99 ± 0.08 g kg⁻¹ and -0.24 ± 0.07 ‰ between 46°S and 60°S (Supplementary Fig. 1). There is no reason to believe that the accuracy for surface salinity and $\delta^{18}O$ is lower in the northern region than in the southern region. Instead, this variability is likely due to the presence of the energetic Antarctic Circumpolar Current, with the presence of eddies and large transient variability. Such

variability superimposes on and ‘hides’, to some degree, the long-term trends. This is reflected in a larger error envelop when we compute the long-term trend, and ultimately this is propagated in the flux change estimates.

Line 149-151: Equations 4 and 5 depend on the mean isotopic values and the the error ranges, are these ranges taken into account in the boot-strapping solutions in surface salinity error bars?

Yes, the error ranges on the mean isotopic values are indeed taken into account in the bootstrapping. We used estimates and their associated errors from the literature and reported those ranges in Table 1. From Eqs 4 and 5, we then select all solutions of the mean isotopic composition of precipitation in the northern and the southern sectors that allow us to explain the observed salinity and $\delta^{18}\text{O}$ trends. We made an attempt to clarify the main text as follows. Line 171:

‘Our multidecadal trend estimates are based on a mean isotopic composition of precipitation of $-7 \pm 2\text{‰}$ in the northern sector and $-11 \pm 3\text{‰}$ in the southern sector, consistent with previously reported estimates and their associated errors⁴² reported in Table 1 and taken into account in the equation solutions.’

Line 159-160: “.. leading to a net surface ocean freshening... “, I think it needs to be made clear that this ocean freshening is for the region where the sea-ice is transported too, rather than say the coastal or polynya regions where the sea-ice is formed “surface salinity” would increase from this contribution.

Accepted. We amended the main text. Line 180:

‘In contrast, sea ice decline induces a positive salinity trend ($0.008 \pm 0.002 \text{ g kg}^{-1}$ per decade) in the northern part of the sea ice sector and further north in the subpolar Indian sector of the Southern Ocean. This positive trend is compensated and surpassed by a negative trend induced by P-E ($-0.02 \pm 0.008 \text{ g kg}^{-1}$ per decade), leading to net surface ocean freshening.’

Line 167: “... only a limited influence of an increase in Antarctic discharge on the surface water ...”

We thank the reviewer and we corrected line 197:

‘Second, our findings suggest only a limited influence of an increase in Antarctic ice discharge on the surface water properties in the subpolar ocean, away from the continent.’

Line 176-176: This last sentence is at odds with the earlier assertion that the larger scale surface ocean salinity map and the changes described in this region is consistent, and here they should not be extrapolated to the whole southern ocean. And inconsistent with the discussion in the methods section. I think is is useful to provide the back of envelope estimates for the whole of Southern from this sector, keeping in mind these are trends that are derived from data nearly 30 years.

We respectfully disagree with the reviewer here. While the reviewer is correct that the freshening trend of our sector is relatively consistent across the entire Southern Ocean, the key aspect of our study is the attribution of this freshening trend. The freshening can come from diverse processes as explained in our paper, and this is likely region dependent. Arguably the ice sheet component would become a bigger player closer to the continental shelf. Sea ice might have an opposite contribution depending on the sector. At least, there is no reason to believe that processes explaining freshening should be consistent across the Southern Ocean. We therefore prefer not providing an extrapolation of our results.

Couple of specific comments: the paper reports the average surface salinity changes. But actually what is changing is the depth integrated mixed layer salt content, and P-E is reported as g/kg per decade. So it would be helpful to have a table of the more common flux changes, so P-E would be mm/year per decade. This would allow a comparison with the NCEP or ERA P-E values. In this work I have not seen a table or the equivalent P-E values from NCEP (I note the discussion in methods) compared with the results from this work. The inhomogeneities and inconsistencies become a case that support these measurements. Such a table would combine the re-analyses products, the ice melt and the sea-ice transport values altogether.

We thank the reviewer; we added a table of the different P-E flux trends estimated in the supplementary information (see Supplementary Note 2 and Supplementary Table 1).

There is no discussion of the trends of temperature in the measurements. Has the mixed layer thinned and so these freshening values in the southern ocean are a result of a thinner mixed layer. Its unclear whether the mixed layer depth is variable in space (in the salinity balance equations).

We thank the reviewer for this comment. We only focus on the salinity and $\delta^{18}\text{O}$ of surface waters here as the objective is to investigate the contribution of the freshwater fluxes to the surface properties changes. To this aim, the surface temperature is not required as additional information for this paper. As mentioned above in a previous response to the reviewer, the mixed layer has only slightly deepened during that period in this region according to Sallée et al., 2021. As the mixed layer deepened since 1970, the observed freshening is unlikely driven by mixed layer depth changes but rather an amplification of the hydrological cycle with contribution of glacial meltwater input and sea ice changes.

Figure 1. Last sentence in figure caption. I wondered whether the black line was being used.

We removed the black line as we agree that this information is not pertinent for the present study.

Figure 3. Looking in these error bars I wondered how robust the trends are in the trends, and if one was to remove a few values from each panel, whether the estimates of the trends would be as stable as suggested by the linear trends with associated errors. There are relatively few degrees of freedom, and looking at the seasonal values, there is some serial correlation. It would be good to test the reliability of the trends here by subsetting the values to see if the estimates are stable, and then say so.

We understand the concern of the reviewer regarding the trend estimates from a sparse dataset. We actually do apply a bootstrapping method in which we sub set the values, as suggested by the reviewer. In this endeavor which aims at measuring the errors of our trend estimate, we repeat the same calculation many times, but sub setting only 80 % of the values randomly at each iteration. We hope that applying bootstrapping to the trend calculations in the main text, the reviewer will be convinced about the robustness of the trends.

References

Chapman et al. "Defining Southern Ocean fronts and their influence on biological and physical processes in a changing climate." *Nature Climate Change* 10.3 (2020): 209-219.

Meredith et al. "Polar Regions. Chapter 3, IPCC Special Report on the Ocean and Cryosphere in a Changing Climate." (2019).

Fox-Kemper et al. "Ocean, Cryosphere and Sea Level Change. In *Climate Change 2021: The Physical Science Basis. Contribution of Working Group I to the Sixth Assessment Report of the Intergovernmental Panel on Climate Change.*" (2021) Cambridge University Press, Cambridge, United Kingdom and New York, NY, USA, pp. 1211–1362

Turner et al. "Unprecedented springtime retreat of Antarctic sea ice in 2016." *Geophysical Research Letters* 44.13 (2017): 6868-6875.

Meredith et al. "Tracing the impacts of recent rapid sea ice changes and the A68 megaberg on the surface freshwater balance of the Weddell and Scotia Seas". Accepted for publication in *Philosophical Transactions of the Royal Society A* (2023).

Fogt et al. "A regime shift in seasonal total Antarctic sea ice extent in the twentieth century." *Nature Climate Change* 12.1 (2022): 54-62.

Jacobs et al. "Freshening of the Ross Sea during the late 20th century." *science* 297.5580 (2002): 386-389.

Akhoudas et al. "Ice shelf basal melt and influence on dense water outflow in the Southern Weddell Sea." *Journal of Geophysical Research: Oceans* 125.2 (2020): e2019JC015710.

Akhoudas et al. "Ventilation of the abyss in the Atlantic sector of the Southern Ocean." *Scientific Reports* 11.1 (2021): 6760.

Morrow and Kestenare. "Nineteen-year changes in surface salinity in the Southern Ocean south of Australia." *Journal of Marine Systems* 129 (2014): 472-483.

Adusumilli et al. "Interannual variations in meltwater input to the Southern Ocean from Antarctic ice shelves." *Nature geoscience* 13.9 (2020): 616-620.

Greene et al. "Antarctic calving loss rivals ice-shelf thinning." *Nature* 609.7929 (2022): 948-953.

Speer et al. "Antarctic Mode Water." *Journal of Marine Research* 76.3-4 (2018): 119-137.

Cerovečki et al. "Subantarctic mode water formation, destruction, and export in the eddy-permitting Southern Ocean state estimate." *Journal of physical oceanography* 43.7 (2013): 1485-1511.

Haumann et al. "Sea-ice transport driving Southern Ocean salinity and its recent trends." *Nature* 537.7618 (2016): 89-92.

Sallée et al. "Summertime increases in upper-ocean stratification and mixed-layer depth." *Nature* 591.7851 (2021): 592-598.

Reviewer #2 (Remarks to the Author):

The authors have nicely addressed my comments. I recommend this manuscript to be published in Nature Communications as it provides compelling evidence of changes in the hydrological cycle in the Southern Ocean using innovative applications of seawater isotopes.

Reviewer #3 (Remarks to the Author):

Review of "Isotopic evidence for an intensified hydrological cycle in the Indian sector of the Southern Ocean" Camille Hayatte³, Akhoudas^{1,2*}, Jean-Baptiste Salle³, Gilles Reverdin³, F. Alexander⁴, Haumann^{4,5,6}, Etienne Pauthenet⁷, Christopher C. Chapman⁸, Felix Margirier⁹, Claire Lo Monaco³, Nicolas Metz¹³, Julie Meilland¹⁰, and Christian Stranne^{1,2}

General Comments:

The paper is much revised.

I liked the additional materials in the supplementary materials on the estimate of sea-ice melt and also the comparison of the three reanalysis products, and their similarities and differences. While these results are not exactly what I would have anticipated they make the case that the oxygen isotopes do have a lot to offer in terms of making estimates of the strength of the P-E to the north of 46S and the region to the south of 46S.

The additional text around the methods and the solution of the 1-D freshwater balances and the a priori estimate of freshwater from sea-ice is much clearer now. The response to my comments are satisfactory except where I note below. I have some additional comments below about strengthening the message of this paper (in abstract and in the main text).

The recognition of the mixed layer, and better discussion of the salinity balance.

The figures are clear, and the written text is of high quality. The supplementary materials are complete and appropriate.

I believe the manuscript is acceptable for publication.

Comments below on this revision.

Abstract:

Line 18, middle sentence, "... decreasing it in subpolar surface waters ..." reads better I think "...decreasing salinity in subpolar surface waters by ..." At first I thought it referred to the atmospheric water cycle.

There is a common misconception that all of the salinity changes are caused by melt from Antarctica, so I suggest that the last sentence, could be augmented by inserting the following sentence before the last sentence with some like "The oxygen isotope data show that the freshening in in subpolar waters is largely driven by the increase in P-E (by a factor of two) while the decrease in sea-ice melt is largely balanced by the contribution glacial melt water at these latitudes. These changes..." The message here could be equally well accomplished by re-phrasing the text in the current second last sentence. To me the power and success of this paper is the use of the oxygen isotopes and the power they give to discriminate the different freshwater processes.

The text throughout has error bars, but not in the abstract, perhaps add.

Main text:

Specific comments.

Line 64: To be clear "... local anomaly ..." should be a "... local spatial anomaly..."

Line 131 to 133. Suggest "...per decade, which is a relative change that would be more than ..." to "... per decade, and this deepening represents a third of the relative change in P-E based on the atmospheric reanalysis estimates (ERA5 ..."

Line 134 to 154: In all of these estimates the seasonal cycle is not mentioned, and yet later in the text under methods the seasonal cycle is discussed in the integration of the freshwater fluxes (line 226). It would be great to include an explicit statement that the seasonal cycle has been taken into account by integrating over the season or by saying that the estimates presented are the annual averages (thus taking into account the seasonal cycle).

Line 181: The sea-ice decline is really a statement of less sea-ice melt in this zone compared with earlier times, and since extent has not changed it really means that the amount ice formed nearer to the continent has actually reduced and the ice has thinned (overall), ie volume of sea-ice is declining. I suggest bring this point out more firmly. "... sea ice decline (implying less formation and volume) induces ..."

Line 209 to 215: In the first revision I noted a "logical inconsistency between the acceleration of the hydrological cycle, which is more a global parameter, revealed more through zonal averages. So perhaps not drawing to negative argument, that the results are only applicable to this region, and not elsewhere - which raises larger questions about spatial and temporal variations and uncertainty in that context, it would be better extol the virtues of the oxygen isotopes to pull apart the various processes and thus provide the first quantifiable evidence of the strength of the different freshwater processes in this region, and then extend to the larger region. I would reframe this way. At the moment it undermines the great results you have collectively for this region.

We would like to thank the reviewers for this second round of review which provide constructive suggestions to improve our manuscript. Our responses to comments are below in blue and line numbers correspond to the new version of the manuscript.

Reviewer #2 (Remarks to the Author):

The authors have nicely addressed my comments. I recommend this manuscript to be published in Nature Communications as it provides compelling evidence of changes in the hydrological cycle in the Southern Ocean using innovative applications of seawater isotopes.

We thank the reviewer for this careful reading of the revised version of the manuscript and our responses to the comments. We are very pleased that the reviewer is convinced by our work and recommend our manuscript to be published in Nature Communications.

Reviewer #3 (Remarks to the Author):

Review of “Isotopic evidence for an intensified hydrological cycle in the Indian sector of the Southern Ocean” Camille Hayatte³, Akhoudas^{1,2*}, Jean-Baptiste Salle³, Gilles Reverdin³, F. Alexander⁴, Haumann^{4,5,6}, Etienne Pauthenet⁷, Christopher C. Chapman⁸, Fe³lix Margirier⁹, Claire Lo Monaco³, Nicolas Metz¹³, Julie Meilland¹⁰, and Christian Stranne^{1,2}

General Comments:

The paper is much revised.

I liked the additional materials in the supplementary materials on the estimate of sea-ice melt and also the comparison of the three reanalysis products, and their similarities and differences. While these results are not exactly what I would have anticipated they make the case that the oxygen isotopes do have a lot to offer in terms of making estimates of the strength of the P-E to the north of 46S and the region to the south of 46S.

The additional text around the methods and the solution of the 1-D freshwater balances and the apriori estimate of freshwater from sea-ice is much clearer now. The response to my comments are satisfactory except where I note below. I have some additional comments below about strengthening the message of this paper (in abstract and in the main text).

The recognition of the mixed layer, and better discussion of the salinity balance.

The figures are clear, and the written text is of high quality. The supplementary materials are complete and appropriate.

I believe the manuscript is acceptable for publication.

We thank the reviewer for this careful reading of the revised version of the manuscript and for these positive comments. The suggestions provided for this last round of review have been addressed in the manuscript and detailed below. We are very pleased that the reviewer is convinced by our work and recommend our manuscript to be published in Nature Communications.

Comments below on this revision.

Abstract:

Line 18, middle sentence, “.. decreasing it in subpolar surface waters ...’ reads better I think

“...decreasing salinity in subpolar surface waters by ...” At first I thought it referred to the atmospheric water cycle.

There is a common misconception that all of the salinity changes are caused by melt from Antarctica, so I suggest that the last sentence, could be augmented by inserting the following sentence before the last sentence with some like “The oxygen isotope data show that the freshening in in subpolar waters is largely driven by the increase in P-E (by a factor of two) while the decrease in sea-ice melt is largely balanced by the contribution glacial melt water at these latitudes. These changes... “ The message here could be equally well accomplished by re-phrasing the text in the current second last sentence. To me the power and success of this paper is the use of the oxygen isotopes and the power they give to discriminate the different freshwater processes.

The text throughout has error bars, but not in the abstract, perhaps add.

We thank the reviewer; we re-phrased the abstract based on the suggestions made above:

‘The hydrological cycle is expected to intensify in a warming climate. However, observational evidence of such changes in the Southern Ocean is difficult to obtain due to sparse measurements and a complex superposition of changes in precipitation, sea ice, and glacial meltwater. Here we disentangle these signals using a dataset of salinity and seawater oxygen isotope observations collected in the Indian sector of the Southern Ocean. Our results show that the atmospheric water cycle has intensified in this region between 1993 and 2021, increasing the salinity in subtropical surface waters by $0.06 \pm 0.07 \text{ g kg}^{-1}$ per decade, and decreasing the salinity in subpolar surface waters by $-0.02 \pm 0.01 \text{ g kg}^{-1}$ per decade. The oxygen isotope data allow to discriminate the different freshwater processes showing that in the subpolar region, the freshening is largely driven by the increase in net precipitation (by a factor two) while the decrease in sea ice melt is largely balanced by the contribution of glacial meltwater at these latitudes. These changes extend the growing evidence for an acceleration of the hydrological cycle and a melting cryosphere that can be expected from global warming.’

Main text:

Specific comments.

Line 64: To be clear “... local anomaly ..” should be a “... local spatial anomaly...”

We thank the reviewer and this was corrected on lines 65-66:

‘Based on the zonally averaged meridional profiles, we define the local spatial anomaly for each individual observation with respect to the climatology in Fig. 2a,b.’

Line 131 to 133. Suggest “...per decade, which is a relative change that would be more than ...” to “... per decade, and this deepening represents a third of the relative change in P-E based on the atmospheric reanalysis estimates (ERA5 ...”

We thank the reviewer and this was corrected on lines 132-134:

‘Ref. ¹² showed that in this region the summer mixed layer depth has deepened at a rate of $2 \pm 4\%$ per decade, and this deepening represents a third of the relative change in P-E based on the atmospheric reanalysis estimates (ERA5, JRA55 and GPCP/OAFlux)’

Line 134 to 154: In all of these estimates the seasonal cycle is not mentioned, and yet later in the text under methods the seasonal cycle is discussed in the integration of the freshwater fluxes (line 226). It

would be great to include an explicit statement that the seasonal cycle has been taken into account by integrating over the season or by saying that the estimates presented are the annual averages (thus taking into account the seasonal cycle).

We thank the reviewer. We have clarified in the text, as early as in the introduction that our analysis only covers summer months because of data availability, consistent with what is discussed in the methodology section. Lines 37-39:

‘In this paper, we explore long-term observations of seawater oxygen isotope and salinity in surface waters of the Indian sector (between 40°E and 90°E, Fig. 1a,b) of the Southern Ocean to document regional ocean changes, and explore potential causes of surface freshening. Our analysis focuses only on the summer season, when observations are available.’

We also clarified in the discussion. Lines 159-161:

‘Here, in addition to salinity, we use $\delta^{18}\text{O}$ observations and focus on surface water properties in the region of the Indian sector of the Southern Ocean between 1993 and 2021 during summer months (December to February).’

Line 181: The sea-ice decline is really a statement of less sea-ice melt in this zone compared with earlier times, and since extent has not changed it really means that the amount ice formed nearer to the continent has actually reduced and the ice has thinned (overall), ie volume of sea-ice is declining. I suggest bring this point out more firmly. “... sea ice decline (implying less formation and volume) induces ...”

We thank the reviewer and this was corrected on lines 182-184:

‘In contrast, sea ice decline (implying less formation and volume) induces a positive salinity trend ($0.008 \pm 0.002 \text{ g kg}^{-1}$ per decade) in the northern part of the sea ice sector and further north in the subpolar Indian sector of the Southern Ocean.’

Line 209 to 215: In the first revision I noted a “logical inconsistency between the acceleration of the hydrological cycle, which is more a global parameter, revealed more through zonal averages. So perhaps not drawing to negative argument, that the results are only applicable to this region, and not elsewhere - which raises larger questions about spatial and temporal variations and uncertainty in that context, it would be better extol the virtues of the oxygen isotopes to pull apart the various processes and thus provide the first quantifiable evidence of the strength of the different freshwater processes in this region, and then extend to the larger region. I would reframe this way. At the moment it undermines the great results you have collectively for this region.

We agree with this comment and thank the reviewer for this suggestion. We changed the message of the last paragraph of the discussion section in order to extol the strength of the study based on oxygen isotopes data allowing us to disentangle the processes at play here. Lines 210-215:

‘Finally, our results provide the quantifiable evidence of the contribution of different freshwater fluxes in the Indian sector of the Southern Ocean (40-90°E and 30-60°S. The use of the approach here whose strength comes from the concurrent observations of oxygen isotopes and salinity to disentangle the processes at play in surface salinity changes should be replicated in other regions of the Southern Ocean to address the spatially complex pattern of changes in surface water fluxes and their imprint on the ocean^{5,54}. The different responses of local regime to direct or remote forcing are important to consider in order to investigate surface changes in the Southern Ocean as a whole.’